# *Trichomonas vaginalis* extracellular vesicles activate the NLRP3 inflammasome and TLR3-mediated inflammatory cascades in host cells

Ching-Yun Huang[1,2], Shu-Fang Chiu[1,2,3], Lichieh Julie Chu[4,5,6], Yuan-Ming Yeh[7], Po-Jung Huang[7,8], Kuo-Lun Lan[9], Yu-Ling Tsai[9], Ching-Chun Liu[2,10], Chien-Fu F. Chen[11], Wei-Ning Lin[12], Kuo-Yang Huang [ID][2,10]*

**1** Graduate Institute of Medical Sciences, National Defense Medical Center, Taipei city, Taiwan, **2** Host-Parasite Interactions Laboratory, National Defense Medical Center, Taipei city, Taiwan, **3** Department of Inspection, Taipei City Hospital, Taipei City, Taiwan, **4** Graduate Institute of Biomedical Sciences, Chang Gung University, Taoyuan, Taiwan, **5** Molecular Medicine Research Center, Chang Gung University, Taoyuan, Taiwan, **6** Department of Otolaryngology - Head & Neck Surgery, Chang Gung Memorial Hospital, Linkou, Taoyuan, Taiwan, **7** Genomic Medicine Core Laboratory, Chang Gung Memorial Hospital, Taoyuan, Taiwan, **8** Department of Biomedical Sciences, Chang Gung University, Taoyuan, Taiwan, **9** Department of Pathology, Tri-Service General Hospital, National Defense Medical Center, Taipei city, Taiwan, **10** Graduate Institute of Pathology and Parasitology, National Defense Medical Center, Taipei City, Taiwan, **11** Graduate Institute of Life Sciences, National Defense Medical Center, Taipei city, Taiwan, **12** Graduate Institute of Biomedical and Pharmaceutical Science, Fu Jen Catholic University, New Taipei City, Taiwan

* cguhgy6934@gmail.com

## Abstract

*Trichomonas vaginalis* (TV) is a flagellated parasite that causes trichomoniasis, the most common non-viral sexually transmitted infection (STI), with over 275 million cases annually. TV has been shown to secrete extracellular vesicles (TV-EVs) to regulate intercellular communication between parasites and host immune response; however, the mechanisms by innate immunity against TV-EVs are largely unknown. Herein, we aim to investigate the molecular mechanisms of inflammation induced by TV-EVs and identify novel proteins modulating the immune response in host cells. Firstly, the morphological characteristics of TV-EVs have been analyzed by transmission electron microscope (TEM) and nanoparticle tracking analysis, revealing that the vesicles are round-shaped bilayer membrane structures with size mostly about 100–120 nm. Additionally, the internalization of TV-EVs by host cells has been validated through immunofluorescence and TEM analysis. The multiplex immunoassay identified that TV-EVs induce the secretion of inflammatory cytokines, including CXCL1, IL-6, IL-8 and MIP-1β in THP-1 macrophages and ectocervical cells (Ect). Mechanistically, TV-EVs induce TLR3 overexpression to activate the NF-κB/NLRP3 pathway in THP-1 macrophages. Additionally, TV-EVs activate the PI3K-mediated NF-κB, p38 MAPK and ERK pathways in Ect. Moreover, TV-EV-induced TLR3 overexpression positively regulates the PI3K and NF-κB pathways, while simultaneously suppressing the p38 MAPK and ERK pathways in Ect. Proteomic analysis identified that TV-EVs

**Data availability statement:** All relevant data are within the manuscript and its Supporting Information files.

**Funding:** This study was supported by grants from the National Science and Technology Council (MOST 110-2320-B-016-011-MY3 and 113-2320-B-016-006 to KYH; 111-2320-B-182-030 and 112-2320-B-182-015 to LJC), National Defense Medical Center (MND-MAB-D-111106 and MND-MAB-D-112099 to KYH), Taipei City Hospital, Renai Branch [TPCH-112-14 to SFC], Department of Health, Taipei City Government [111F210203R24F353 to SFC], and Proteomics Core Lab and Bioinformatics Core Laboratory, Molecular Medicine Research Center, Chang Gung University, Taiwan [grant CLRPD1J0015 to LJC]. The funders had no role in study design, data collection and analysis, decision to publish, or preparation of the manuscript.

**Competing interests:** The authors have declared that no competing interests exist.

upregulate MICB and TRAF3IP2, which are also positively regulated by TLR3 and involved in TV-EV-induced inflammatory cascade. Altogether, this study significantly advances our understanding of the immunomodulatory roles of TV-EVs in host cells, paving the way for future treatment of trichomoniasis and TV-associated STIs.

## Author summary

The role of protozoa-derived extracellular vesicles (EVs) in host-parasite communication may be pivotal for understanding the complexity of their pathogenic mechanisms. *Trichomonas vaginalis* (TV) has been shown to secrete EVs (TV-EVs) to regulate the host immune response; however, the immunomodulatory mechanisms of TV-EVs in host cells remain poorly understood. In this study, we demonstrated that TV-EVs induce TLR3 overexpression, leading to the activation of the NF-κB/NLRP3 pathway in THP-1 macrophages. Additionally, TV-EVs activate the PI3K-mediated NF-κB, p38 MAPK, and ERK pathways in ectocervical cells. Using a proteomic approach, we identified novel proteins, MICB and TRAF3IP2, which are positively regulated by TLR3 and involved in TV-EV-induced inflammation. Our study reveals the biological significance of the inflammatory cascade induced by TV-EVs, which may serve as potential therapeutic targets for trichomoniasis and TV-associated sexually transmitted infections.

## Introduction

*Trichomonas vaginalis* (TV) is a flagellated protozoan parasite that colonizes the urogenital tract of women and men [1] and is commonly transmitted through sexual intercourse. TV is the etiologic agent of trichomoniasis, the most common non-viral sexually transmitted infection (STI), infecting over 275 million people annually worldwide [2]. Common symptoms of TV infection range from asymptomatic infections to vaginitis, urethritis, Pelvic Inflammation Disease (PID) [3,4], prostatitis, and low birth weight [5]. Additionally, it may increase the risk of contracting human immunodeficiency virus (HIV), cervical cancer, and prostate cancer [6,7]. Previous studies have shown that TV attaches to vaginal epithelial cells and induces inflammation in the vagina [8,9]. However, the molecular mechanism of inflammation leading to trichomoniasis is far from understood.

TV infection has been shown to result in the secretion of the inflammatory cytokine interleukin 1 beta (IL-1β), which is dependent upon the nucleotide-binding oligomerization domain leucine-rich repeat and pyrin domain containing 3 inflammasome (NLRP3) activation in prostate epithelial cells [10]. Additionally, TV can induce NLRP3 inflammasome activation, pyroptotic cell death, and IL-1β secretion in human macrophages [11]. Toll-like receptors (TLR) are a class of membrane receptors with an extracellular region that senses different microbes, triggering the activation of anti-pathogen signaling [12,13], and ultimately leading to the activation

of downstream inflammatory factors [14]. For instance, TV infection can activate TLR4 in the genital tract, as evidenced by analysis of cervicovaginal lavage [12]. Additionally, TV infection induces the release of IL-8 and tumor necrosis factor α (TNF-α) by upregulating the mRNA expression of TLR2, TLR4, and TLR9 through the p38-mitogen-activated protein kinase (p38 MAPK) signaling pathway in HeLa cells [15]. Moreover, TV triggers the secretion of pro-inflammatory cytokines, including IL-6, interferon-gamma (IFN-γ), and TNF-α, through the TLR2-mediated p38 MAPK and extracellular signal-regulated kinase (ERK) signaling pathways in mouse macrophages [16]. These findings indicate that TV infection results in the production of pro-inflammatory cytokines through various TLRs and activation of several inflammatory signaling pathways.

Extracellular vesicles (EVs) are structures with phospholipid bilayer-bound membranes that are released by a variety of human cells. EVs play a crucial role in intercellular communication by delivering cargo to recipient cells [17]. There are three main types of EVs, including exosomes, microvesicles, and apoptotic bodies. Exosomes, also known as intraluminal vesicles (ILVs), are formed by the fusion of multivesicular body (MVB) with the plasma membrane [18], with a particle size ranging from 40 to 120 nm [19]. Microvesicles are directly released by budding or pinching from the cell membrane, with a particle size ranging from 100 nm to 1 μm. Apoptotic bodies are vesicles with a size ranging from 500 nm to 2 μm that are released by dying cells in the final stage of apoptosis [20]. TV has been reported to release EVs containing signaling proteins, metabolic enzymes, and cytoskeletal proteins, which are internalized by ectocervial cells (Ect) and induce the secretion of inflammatory cytokines IL-6 and IL-8 [21]. Additionally, TV-EVs also facilitate parasite interaction and are involved in the adherence to host cells. Recent research indicates that TV virus (TVV) could be encapsulated within TV-EVs, which are then transmitted to host cells and modulate pro-inflammatory responses, including the production of cytokines such as IL-8, IL-6, IL-1β, Regulated on activation, normal T cell expressed and secreted (RANTES), and the activation of nuclear factor kappa-light-chain-enhancer of activated B cells (NF-κB) in human HaCaT cells [22]. In addition to the presence of viral components in TV-EVs, the tsRNA content is significantly enriched in TVV-positive EVs compared to TVV-negative EVs [22,23]. While TV-EVs have the potential to induce an immune response in host cells, the underlying mechanisms remain unknown.

In the present study, we investigated the molecular mechanisms driving inflammation induced by TV-EVs. Our findings revealed that TV-EVs induce NF-κB-dependent NLRP3 inflammasome activation in THP-1 macrophages and trigger the phosphoinositide 3-kinases (PI3K), p38 MAPK, ERK, and NF-κB signaling pathways in Ect, subsequently resulting in the secretion of inflammatory cytokines, including IL-6, IL-8, C-X-C motif chemokine ligand 1 (CXCL1), and macrophage inflammatory protein-1β (MIP-1β). Specifically, we noted that TLR3 plays a regulatory role in these TV-EV-induced inflammatory cascades in host cells. Furthermore, proteomic analysis identified novel proteins involved in TV-EV-induced inflammatory signaling, providing new insights into the host-parasite interactions mediated by TV-EVs.

## Results

### Isolation and characterization of EVs derived from different TV isolates

To investigate the biological functions of TV-EVs in host cells, vesicles were isolated from the parasite growth media as previously described [24]. We analyzed the protein expression patterns of EVs derived from three different TV isolates, including two cell lines (ATCC 30236 and ATCC 50143) and one clinical strain (NDMC5) (Fig 1A). Distinct protein expression patterns were observed between the cell lines and the clinical strain, suggesting different compositions of EVs may exist among different TV strains. To test whether the mammalian exosomal marker CD63 can be used as a biomarker for TV-EVs, the TV-EVs fractions were detected by western blot analysis using an antibody against CD63 (Fig 1B). It is obvious that CD63 was unable to serve as a biomarker for TV-EVs isolated from the culture medium in the absence of serum, suggesting the distinct compositions in TV-EVs. We used GAPDH as a cytosolic marker, confirming that TV-EVs belonged to excretory secretory products (ESPs). The morphology of TV-EVs derived from all TV isolates was examined

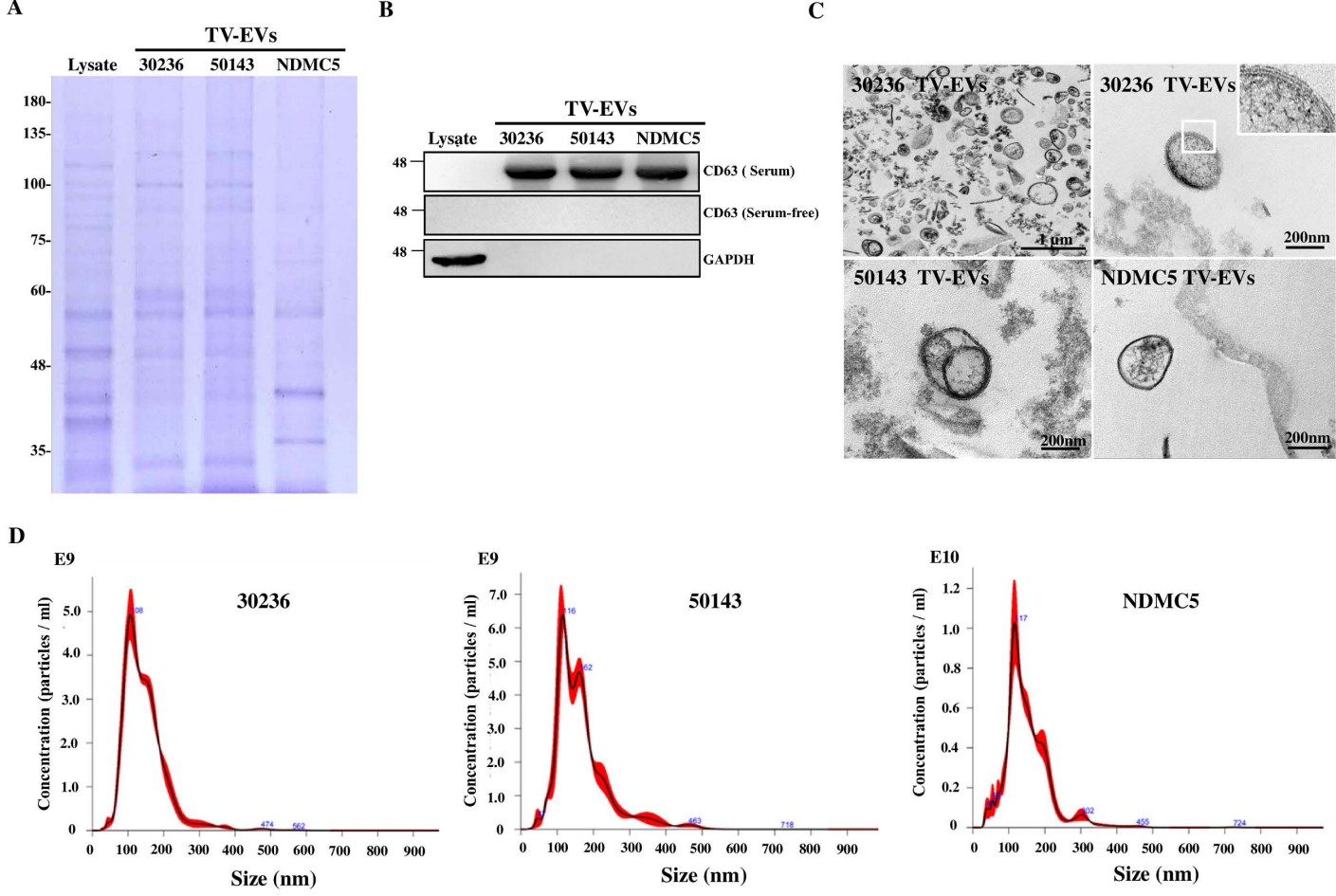

**Fig 1. Physical characterization of TV-EVs purified from different isolates. (A)** Coomassie blue staining was used to analyze the protein compositions of TV cell lysate and TV-EVs isolated from two cell lines (ATCC 50143 and ATCC 30236) and the clinical strain (NDMC5). **(B)** Western blot analysis was utilized to detect the expression of the mammalian exosomal marker CD63 in fractions of TV-EVs isolated from the culture medium in the presence or absence of serum. GAPDH was served as the cytosolic marker. **(C)** TEM was utilized to visualize the ultrastructure of TV-EVs secreted from different isolates. **(D)** NTA was used to determine the concentrations and sizes of EVs derived from different TV isolates.

by transmission electron microscopy (TEM), revealing the presence of round-shaped vesicles with bilayer membranes (Fig 1C). Additionally, nanoparticle tracking analysis (NTA) was employed to further determine the size and concentration of the EVs, indicating a similar particle size of approximately 100–120 nm in different strains (Fig 1D). Together, these results confirm that TV-EVs are rounded, double-membrane vesicles with a size comparable to exosomes.

## Internalization of TV-EVs by THP-1 macrophages and Ect

EVs are able to mediate cell-cell communication by delivering their cargo into host cells. Previous studies reported that *Acanthamoeba castellanii* EVs are internalized by rat glial C6 cells, resulting in increased cytotoxic effects [24]. Similarly, TV-EVs can be taken up by host cells and deliver soluble proteins into Ect [21]. Therefore, we verified whether TV-EVs can be internalized by THP-1 macrophages and Ect. The fluorescent dye PKH67 was used to label the phospholipid bilayer-bound structures of TV-EVs, followed by incubation with THP-1 macrophages and Ect for different time intervals. The cells treated with PBS were served as a negative control. Immunofluorescence data showed that punctate signals of

TV-EVs were detected in host cells after 1 hour of co-incubation and increased in a time-dependent manner (Fig 2A **and 2B**). After 3 hours of co-incubation, numerous punctate signals of TV-EVs were observed fusing with the plasma membranes of THP-1 macrophages and Ect. Additionally, we employed TEM to confirm the internalization of TV-EVs by host cells. Notably, the host cells treated with TV-EVs contained multiple vacuoles that engulfed numerous EV-like vesicles in both THP-1 macrophages (Fig 2C) and Ect (Fig 2D) compared with the PBS-treated control. These data indicate that TV-EVs are internalized by human macrophages and Ect.

To assess the impact of TV-EVs on the viability of host cells, TV-EVs isolated from different strains were incubated with THP-1 macrophages and Ect for different time intervals. The cell morphology of both THP-1 macrophages (S1A Fig) and Ect (S1B Fig) became more irregular and shrinking after 12 hours treatment of TV-EVs. Additionally, the cell viability significantly decreased to approximately 60% in both THP-1 macrophages (S1C Fig) and Ect (S1D Fig) stimulated with TV-EVs for 24 hours. These results indicate that prolonged exposure to TV-EVs leads to a cytotoxic effect in host cells.

## TV-EVs induce inflammatory cytokine production in host cells

TV-EVs have been reported to harbor signaling proteins, metabolic enzymes, and cytoskeletal proteins, inducing inflammatory cytokine secretion [21]. To gain a deeper insight into the impact of TV-EVs on the host immune response, we conducted a multiplex immunoassay to profile 48 common human cytokines in THP-1 macrophages and Ect treated with TV-EVs. The data showed that THP-1 macrophages and Ect co-cultured with TV-EVs purified from different strains induced the secretion of various cytokines (Fig 3A and 3E). Particularly, IL-8, MIP-1β, and CXCL1 exhibited significant increases in THP-1 macrophages treated with TV-EVs for 6 hours compared to the untreated control. Additionally, Ect treated with TV-EVs for 8 hours elicited a significant induction of IL-6, IL-8, and CXCL1 secretion. We used ELISA to validate the production of these inflammatory cytokines in THP-1 macrophages and Ect. Consistent with the screening results, the secretion of IL-8, MIP-1β, and CXCL1 was markedly increased in THP-1 macrophages treated with TV-EVs (P<0.001) (Fig 3B-D), while IL-6, IL-8, and CXCL1 showed significant elevation in Ect treated with TV-EVs (P<0.001) (Fig 3F-H).Collectively, these results indicate that TV-EVs strongly elicit the production of various pro-inflammatory cytokines in host cells.

We further determined whether the release of pro-inflammatory cytokines induced by TV-EVs is observed in vivo. We used an optimized protocol similar to previously established TV infection mouse models [25], which require estrogen treatment combined with the immunosuppressant dexamethasone prior to parasite infection [11]. Mice were inoculated with TV-EVs isolated from the cell line (50143 EV) and the clinical strain (NDMC5 EV) under minimal immune suppression conditions (Fig 3I). The vaginal lavages were collected on days 3, 6, 9, and 12 following the administration of TV-EVs, and the levels of pro-inflammatory cytokines were determined using ELISA. The secretion of IL-8 in mice infected with NDMC5 EVs significantly increased on the third, sixth, and twelfth days. Similarly, the 50143 EV-treated group exhibited a notable elevation in IL-8 secretion on the third and the twelfth days. Additionally, the secretion of CXCL1 was obviously increased in both the NDMC5 EV and 50143 EV groups from the third to the twelfth day (Fig 3J). These results demonstrate that TV-EV treatment leads to increased secretion of pro-inflammatory cytokines in vivo, consistent with our in vitro results.

## TV-EVs activate the NLRP3 inflammasome in THP-1 macrophages

Inflammasomes are multimeric protein complexes that contain inflammatory caspase-1, Nod-like receptors (NLR) family members, and the adaptor apoptosis associated speck like protein containing CARD (ASC) [26,27]. TV has been shown to induce IL-1β release and pyroptosis via activation of the NLRP3 inflammasome in macrophages [11]. Hence, we assessed whether TV-EVs activate the NLRP3 inflammasome to mediate inflammation in THP-1 macrophages. To induce NLRP3 inflammasome activation, we utilized lipopolysaccharide (LPS) and adenosine triphosphate (ATP) as the signal 1 and 2 inducer, respectively [28]. Notably, TV-EVs isolated from different TV strains induced NLRP3 inflammasome

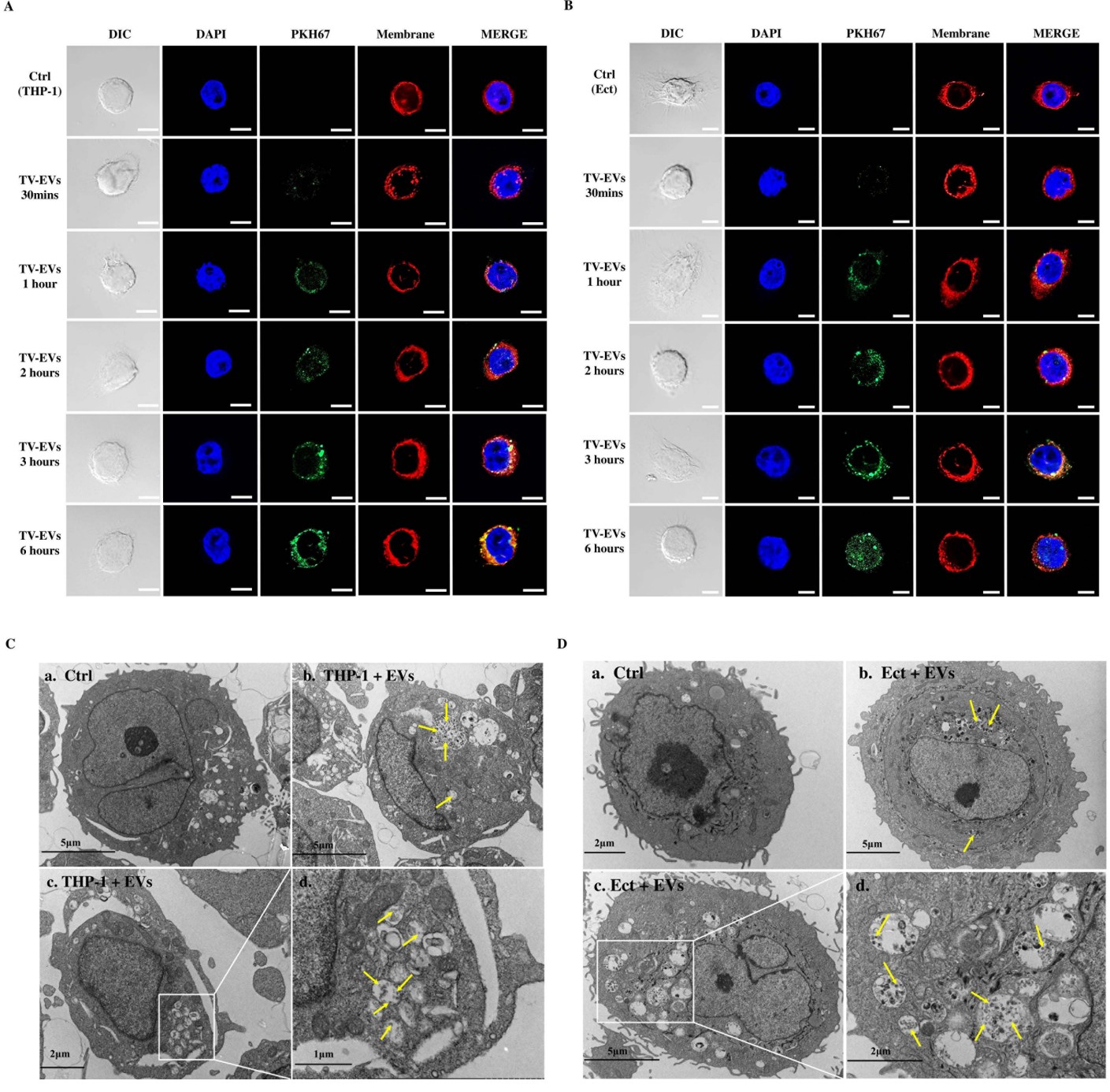

**Fig 2. Internalization of TV-EVs by THP-1 macrophages and Ect. (A)** THP-1 macrophages ($4 \times 10^4$ cells) and **(B)** Ect ($4 \times 10^4$ cells) were co-cultured with PKH67-labeled TV-EVs (30 μg/ml) for different time intervals (30 minutes, 1 hour, 2 hours, 3 hours, and 6 hours). CellMask plasma membrane stain was used to label the plasma membrane, and 4,6-diamidino-2-phenylindole (DAPI) was employed for nuclear staining. Cellular uptake was visualized using a confocal microscope. Scale bar:10 μm. **(C)** $4 \times 10^4$ cells of THP-1 macrophages and **(D)** Ect were co-cultured with TV-EVs (30 μg/ml) or PBS (Control; Ctrl) for 6 hours, and the ultrastructure was determined using TEM. a. PBS-treated control cells (Ctrl), b. and c. THP-1 macrophages or Ect co-cultured with TV-EVs (THP-1 + EVs or Ect + EVs, respectively), d. Enlarged images of boxed areas are shown. Arrows indicate the presence of numerous vesicle-like structures resembling TV-EVs.

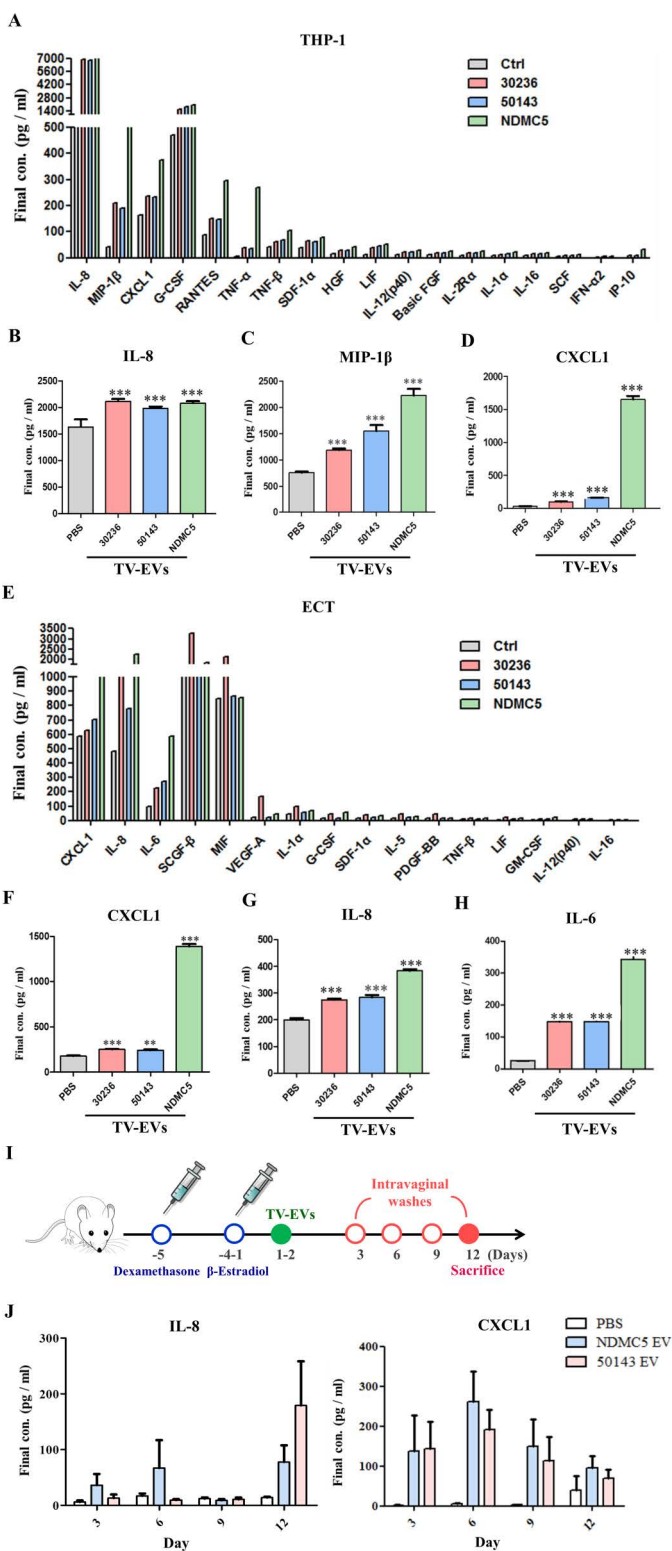

**Fig 3. TV-EVs induce the production of various pro-inflammatory cytokines in the host. (A)** THP-1 macrophages ($2 \times 10^5$ cells/well) were co-cultured with TV-EVs (30 µg/ml) isolated from different strains (ATCC 30236, ATCC 50143, and NDMC5) for 6 hours. The culture supernatants were collected for screening 48 common human cytokines using a multiplex immunoassay. The culture supernatants were assayed for **(B)** IL-8, **(C)** MIP-1β,

and **(D)** CXCL1 using ELISA. **(E)** Ect ($2 \times 10^5$ cells/well) were co-cultured with TV-EVs (30 μg/ml) isolated from different strains for 8 hours and the culture supernatants were collected for screening 48 common human cytokines as mentioned above. The culture supernatants were assayed for **(F)** CXCL1, **(G)** IL-8, and **(H)** IL-6 using ELISA. **(I)** The in vivo model shows the pretreatment strategy for female BALB/c mice before TV-EVs or PBS inoculation. **(J)** Mice vaginas (n = 6 mice per group) were inoculated with TV-EVs isolated from different strains (50143 and NDMC5) for 3, 6, 9, and 12 days and the vaginal lavages were collected and assayed for IL-8 and CXCL1 using ELISA. The quantitative data are expressed as the means ± SEM from three independent experiments. * $p < 0.05$, ** $p < 0.01$, *** $p < 0.001$.

activation in THP-1 macrophages, leading to proteolytic processing of the proenzyme form of caspase-1 (48 kDa) to the active form (p20). Also, NLRP3, ASC, and the mature IL-1β cleavage product (17 kDa) were significantly upregulated upon TV-EV treatment (Fig 4A and 4B). Additionally, IL-1β secretion was remarkably increased in THP-1 macrophages treated with TV-EVs compared with the PBS-treated control (Fig 4C). Moreover, the co-localization of NLRP3 and ASC proteins was markedly increased in THP-1 macrophages after stimulation with TV-EVs (Fig 4D). Collectively, these results indicate that TV-EVs activate the NLRP3 inflammasome and subsequently induce IL-1β secretion in THP-1 macrophages.

### NLRP3 inflammasome activation is mediated by NF-κB in THP-1 macrophages stimulated with TV-EVs

NF-κB represents a family of inducible transcription factors, playing a key role in proliferation, cell death, and NLRP3 inflammasome activation [29,30]. We further investigated whether TV-EVs can regulate the NF-κB signaling pathway to activate the NLRP3 inflammasome. THP-1 macrophages were treated with TV-EVs for different time intervals and the phosphorylation of NF-κB p65 was assessed. Notably, NF-κB p65 activation was observed after 15 minutes of TV-EV treatment (P < 0.001), and this activation persisted for at least four hours (Fig 5A and 5B). Additionally, we examined the distribution of NF-κB p65 in THP-1 macrophages treated with TV-EVs. The TV-EV-treated group displayed remarkable nuclear translocation of NF-κB p65 compared to the PBS-treated control (Fig 5C), suggesting NF-κB activation. To further confirm this observation, nuclear and cytoplasmic extracts from THP-1 macrophages treated with TV-EVs for 4 hours were analyzed, revealing that the expression of NF-κB p65 was upregulated in the nuclear fraction of the TV-EV-treated group (S2 Fig). To validate the role of NF-κB in NLRP3 inflammasome activation, THP-1 macrophages were pretreated with the NF-κB inhibitor BAY 11–7082, and the activation of the NLRP3 inflammasome upon TV-EV treatment was determined. It is obvious that the TV-EV-induced upregulation of NLRP3 and mature IL-1β was significantly decreased (P < 0.05) after BAY 11–7082 treatment (Fig 5D). Additionally, we detected the TV-EV-induced secretion of pro-inflammatory cytokines IL-1β, MIP-1β, IL-8, and CXCL1 from THP-1 macrophages after pretreatment with BAY 11–7082, and observed a remarkble decrease in the levels of these cytokines (P < 0.05)(Fig 5E). These data indicate that TV-EVs derived from different strains activate the NLRP3 inflammasome and modulate the secretion of IL-1β, MIP-1β, IL-8, and CXCL1 through the NF-κB signaling pathway.

### TV-EVs induce pro-inflammatory cytokine secretion through the PI3K-mediated AKT, NF-κB, P38 MAPK, and ERK pathways in Ect

The PI3K, AKT, NF-κB, and MAPK signaling pathways are implicated in various immune responses triggered by pathogens [31,32]. We then assessed whether TV-EVs activate these signaling pathways in Ect. Ect were treated with TV-EVs for different durations (0, 2, 4, 6, 8 and 10 hours), and the cells were collected to analyze the phosphorylation of PI3K, AKT, NF-κB p65, p38 MAPK, and ERK using western blot analysis. As shown in Fig 6, the phosphorylation of PI3K, AKT, NF-κB p65, p38 MAPK, and ERK began to increase at 2 hours and remained consistently elevated until 10 hours compared to the control (P < 0.01) (Fig 6A and 6B). Additionally, Ect stimulated with TV-EVs exhibited a significant increase in the nuclear translocation of NF-κB (Fig 6C). These results indicate that the PI3K, AKT, NF-κB, p38 MAPK, and ERK signaling pathways are activated in Ect in response to TV-EVs. To further explore whether the PI3K signaling pathway serves as an upstream regulator of the AKT, NF-κB, p38 MAPK, and ERK pathways, the PI3K signaling was inhibited using the

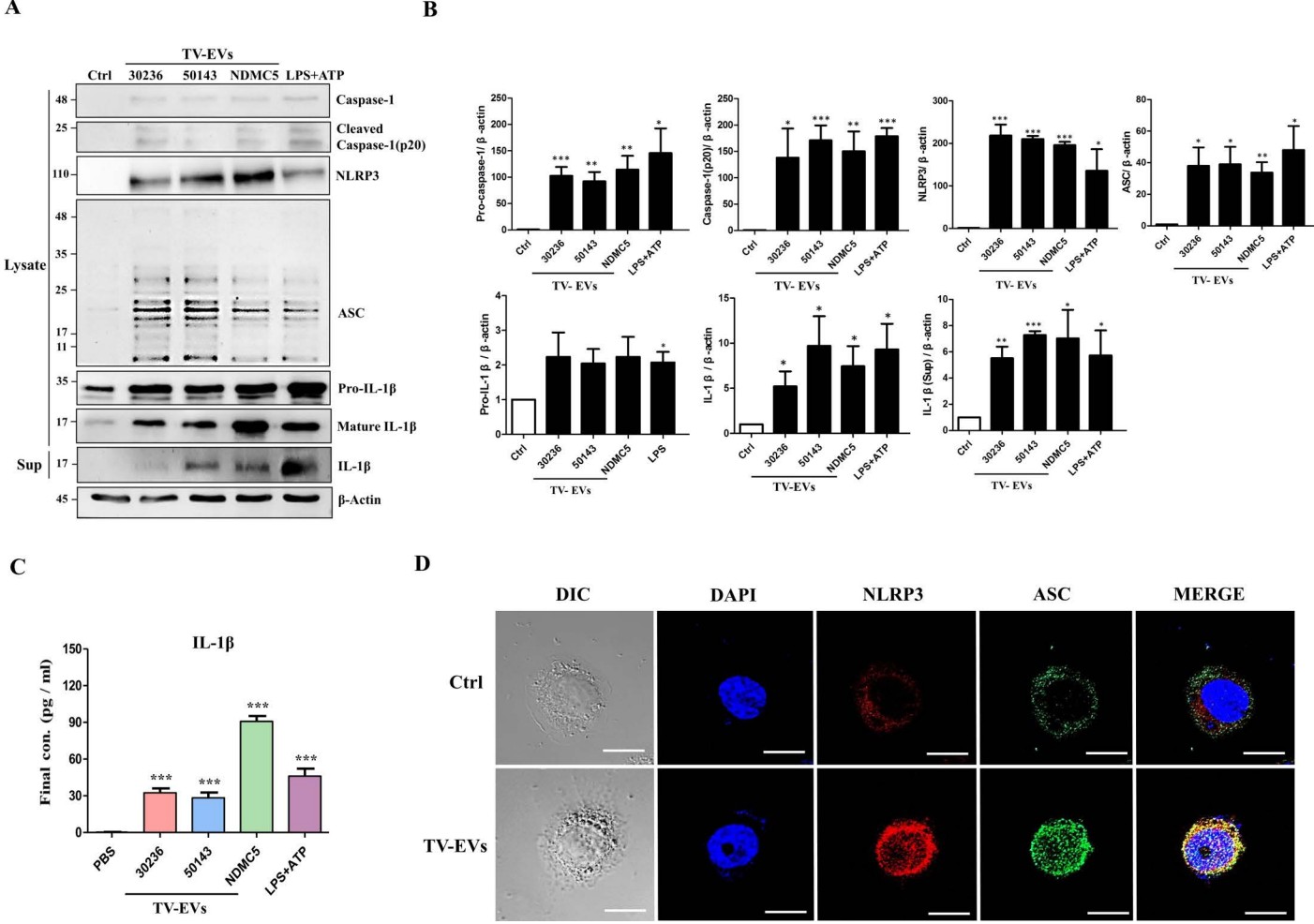

**Fig 4. TV-EVs induce NLRP3 inflammasome activation in THP-1 macrophages.** **(A)** THP-1 macrophages (5 × 10⁶ cells) were treated with the PBS control (Ctrl) or TV-EVs (30 µg/ml) isolated from different strains (ATCC 30236, ATCC 50143, and NDMC5) for 6 hours. Subsequently, the cells were collected (Lysate) and subjected to western blot analysis using anti-NLRP3, anti-caspase-1, anti-ASC, anti-IL-1β, and anti-β-actin antibodies. The supernatant (Sup) was collected for detecting IL-1β secretion. LPS (1 µg/ml) in combination with ATP (30 nM) was used as the positive control for NLRP3 inflammasome activation. The protein expression of β-actin was used as the internal control for western blot analysis. **(B)** Quantification of protein bands from **(A)** was performed using Image J. **(C)** THP-1 macrophages were treated with TV-EVs isolated from different strains or PBS for 6 hours and the supernatants were collected for IL-1β detection using ELISA. **(D)** Immunofluorescence analysis was performed to detect the distribution of NLRP3 and ASC in THP-1 macrophages (4 × 10⁴ cells/ml) treated with TV-EVs or PBS (Ctrl) using antibodies against NLRP3 (red) and ASC (green), followed by visualization under a confocal microscope. Nuclei were stained with DAPI (blue). Scale bar: 10 µm. The data are expressed as the means ± SEM from three independent experiments.* $p < 0.05$, ** $p < 0.01$, *** $p < 0.001$.

PI3K inhibitor wortmannin. The results showed that Ect pretreated with wortmannin significantly inhibited the phosphorylation of PI3K, AKT, p38 MAPK, and ERK (P < 0.05) (Fig 6D), as well as the nuclear translocation of NF-κB p65 (Fig 6E). To verify the involvement of the NF-κB, p38 MAPK, and ERK signaling pathways in the regulation of IL-6, IL-8, and CXCL1 secretion, Ect were pretreated with the NF-κB inhibitor BAY 11–7082, and the two MAPK inhibitors SB203580 (targeting p38 MAPK) and PD98059 (targeting ERK). The secretion of IL-6, IL-8, and CXCL1 induced by TV-EVs was significantly reduced after blocking the NF-κB, p38 MAPK, and ERK pathways (P < 0.05) (Fig 6F). Taken together, our data demonstrate that TV-EVs induce pro-inflammatory cytokine production from Ect through the regulation of PI3K signaling, which, in turn, activates the AKT, NF-κB, p38 MAPK, and ERK pathways.

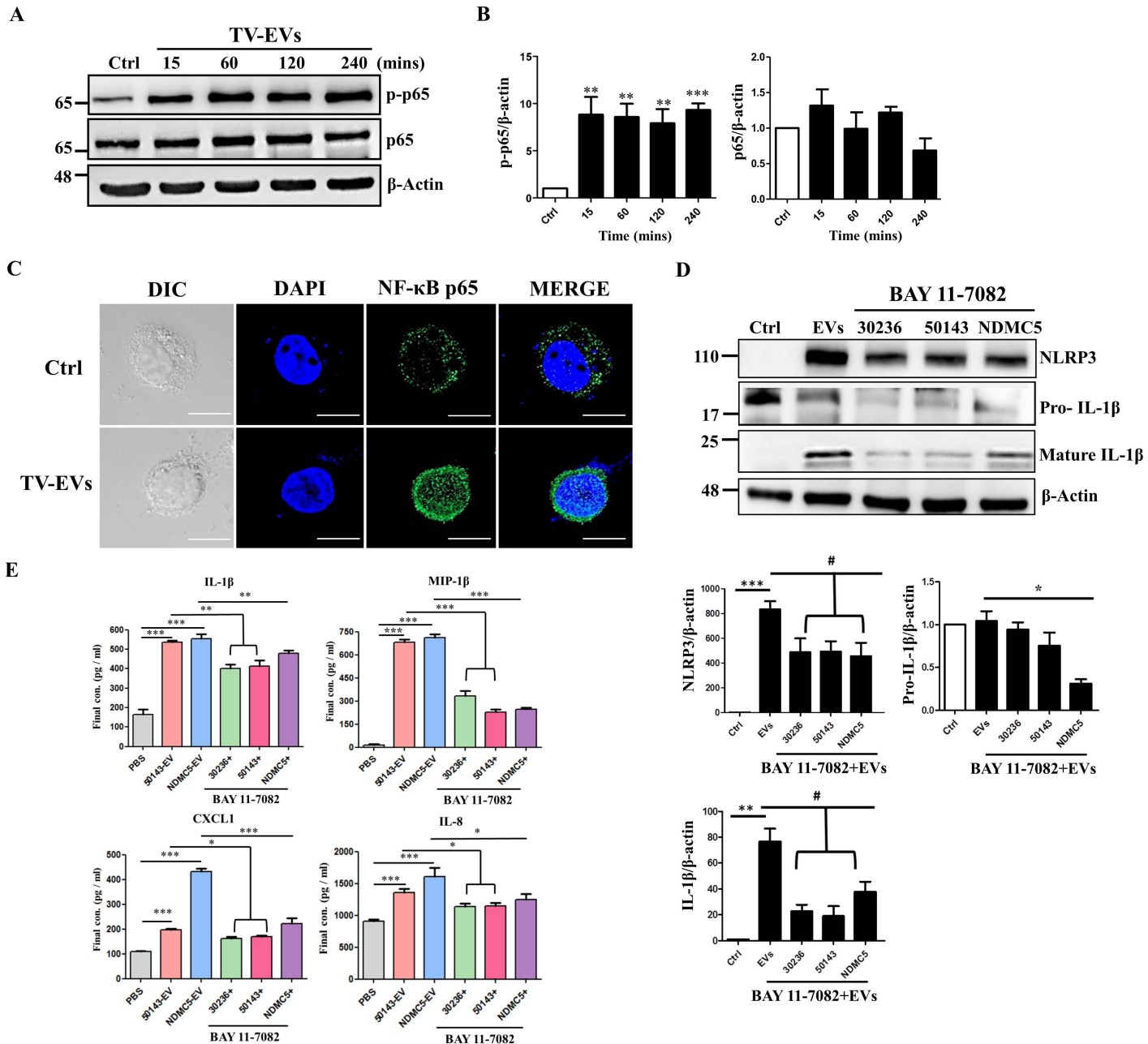

**Fig 5. TV-EVs activate NF-κB signaling to regulate NLRP3 inflammasome activation and pro-inflammatory cytokine secretion in THP-1 macrophages.** **(A)** THP-1 macrophages ($5 \times 10^6$ cells) were treated with the PBS control (Ctrl) or TV-EVs (ATCC 50143) for 15, 60, 120, and 240 minutes. Whole cell lysates were collected and subjected to western blot analysis. The protein expression of NF-κB (p65) and phosphorylated NF-κB (p-p65) was detected using specific antibodies. **(B)** Quantification of protein bands from **(A)** using Image J. **(C)** Immunofluorescence analysis was performed in THP-1 macrophages treated with TV-EVs for 4 hours using an antibody against NF-κB (green), followed by visualization under a confocal microscope. Nuclei were stained with DAPI (blue). **(D)** THP-1 macrophages were pretreated with the NF-κB inhibitor BAY 11–7082, and then co-cultured with TV-EVs isolated from different strains (ATCC 30236, ATCC 50143, and NDMC5) for 6 hours. Whole cell lysates were analyzed by western blot analysis using anti-NLRP3 and anti-IL-1β antibodies. Protein bands were quantified as mentioned above. **(E)** The supernatants collected from THP-1 macrophages ($2 \times 10^5$ cells/well) treated with different conditions as mentioned in **(D)** were analyzed for IL-1β, IL-8, MIP-1β, and CXCL-1 secretion using ELISA. The protein expression of β-actin was used as the internal control for western blot analysis. The quantitative data are expressed as the means ± SEM from three independent experiments. * $p < 0.05$, ** $p < 0.01$, *** $p < 0.001$.

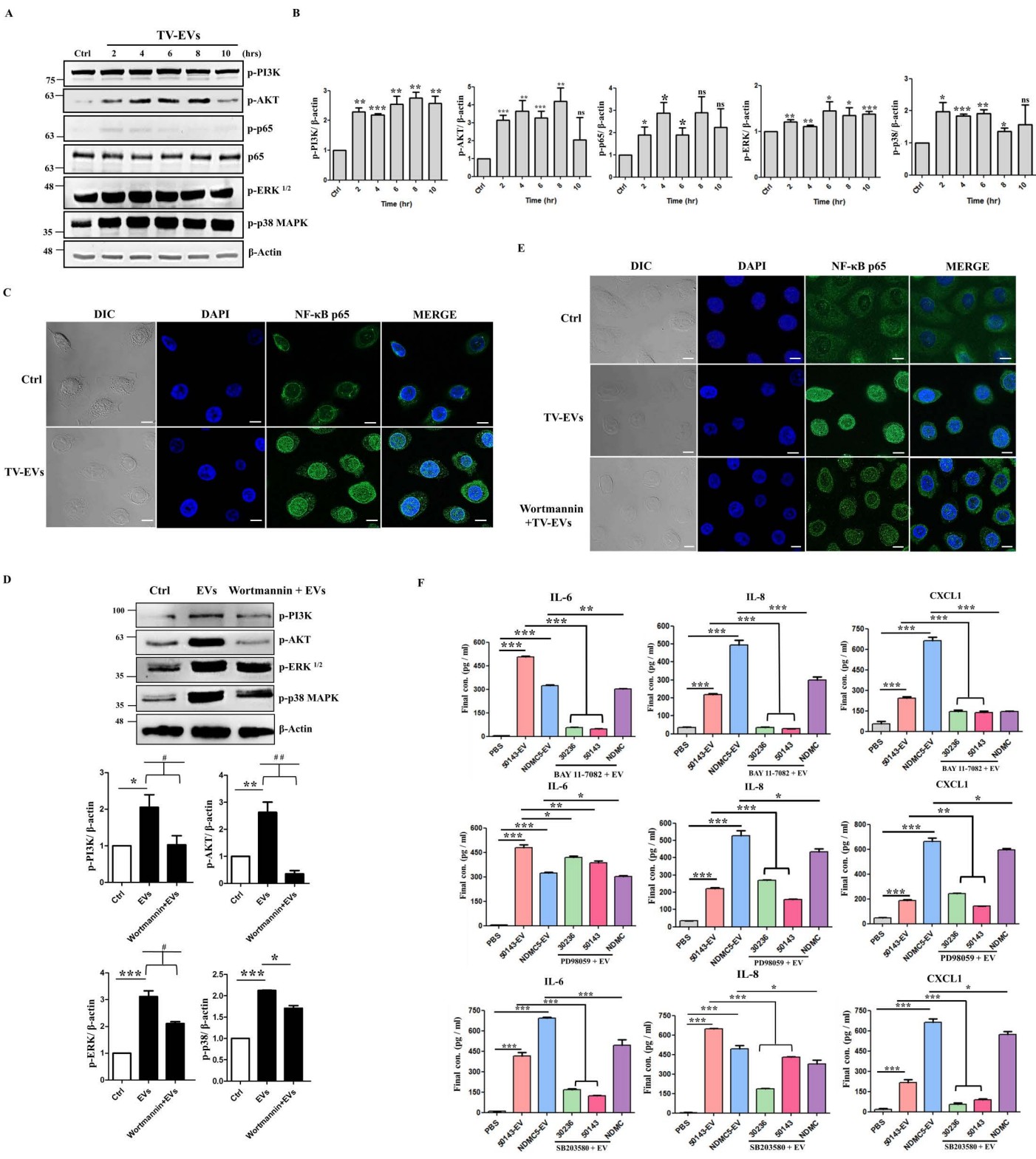

**Fig 6. TV-EVs activate the AKT, NF-κB, p38 MAPK, and ERK pathways through PI3K signaling to mediate inflammation in Ect. (A)** Ect ($2 \times 10^6$ cells) were treated with the PBS control (Ctrl) or TV-EVs (ATCC 50143) for 2, 4, 6, 8, and 10 hours. Subsequently, whole cell lysates were collected and

subjected to western blot analysis. The phosphorylation of PI3K, AKT, NF-κB, p38 MAPK, and ERK was detected using specific antibodies. **(B)** Quantification of protein bands from (A) using Image J. **(C)** Immunofluorescence analysis was performed to detect the localization of NF-κB in Ect treated with TV-EVs compared with the PBS-treated control (Ctrl) using an antibody against NF-κB (green), followed by visualization under a confocal microscope. Nuclei were stained with DAPI (blue). Scale bar: 10 μm. **(D)** Ect were pretreated with the PI3K inhibitor, wortmannin, and then co-cultured with TV-EVs isolated from different strains for 4 hours. The cells were collected and analyzed by western blot analysis using anti-p-PI3K, anti-p-AKT, anti-p-p38 MAPK, and anti-p-ERK antibodies. The protein bands were quantified as mentioned above. **(E)** Ect ($4 \times 10^4$ cells) were pretreated with the PI3K inhibitor wortmannin and then co-cultured with TV-EVs for 4 hours. Immunofluorescence analysis was performed using an anti-NF-κB antibody (green). Nuclei were stained with DAPI (blue). Scale bar:10 μm. **(F)** Ect ($2 \times 10^5$ cells/well) were pretreated with pathway inhibitors, including NF-κB (BAY 11-7082), p38 MAPK (SB203580), or ERK (PD98059), followed by treatment with TV-EVs isolated from different strains (ATCC 30236, ATCC 50143, and NDMC5) for 8 hours. The culture supernatants were collected and assayed for IL-6, IL-8, and CXCL1 by ELISA. Ect treated with the PBS control (Ctrl) and TV-EVs isolated from different strains (50143-EV and NDMC5-EV) were served as a negative and positive control, respectively. The protein expression of β-actin was used as the internal control for western blot analysis. The quantitative data are expressed as the means ± SEM from three independent experiments. * $p < 0.05$, ** $p < 0.01$, *** $p < 0.001$.

## TV-EVs trigger TLR3 overexpression to activate the NF-κB/NLRP3 signaling pathway in THP-1 macrophages

Macrophages play a pivotal role in recognizing pathogen-associated molecular patterns (PAMPs) and damage-associated molecular patterns (DAMPs) through TLRs and NLR, inducing an inflammatory response [13,33]. Mouse macrophages infected with TV have been shown to produce pro-inflammatory cytokines, such as IL-6, TNF-α, and IFN-γ, via the activation of TLR2, which subsequently regulates the p38 MAPK, ERK, and NF-κB signaling pathways [16]. In light of these findings, we initially examined the presence of TLR genes (TLR1–10) in Ect and confirmed the absence of TLR8 and TLR9 gene expression (S3A Fig). We next evaluated the expression levels of TLRs in TV-infected Ect at various time points, revealing a significant increase in gene expression for TLR1, TLR2, TLR3, and TLR5 (S3B Fig). Given the known importance of TLR2 in the context of TV infection, our focus shifted to the protein expression of TLR1, TLR3, and TLR5 in Ect treated with TV-EVs. Remarkably, TLR3 was upregulated in Ect following treatment with TV-EVs (S3C Fig). Hence, we investigated whether TV-EVs induce TLR3 expression to regulate the immune response in THP-1 macrophages. Notably, THP-1 macrophages treated with TV-EVs significantly triggered TLR3 expression within 15 mins (P < 0.05), and this effect persisted for at least 4 hours (Fig 7A). To investigate whether the TV-EV-induced TLR3 upregulation activates the NF-κB/NLRP3 pathway in THP-1 macrophages, the TLR3 inhibitor (614310) and the siRNA targeting TLR3 (si-TLR3) were used to block the function of TLR3. It is obvious that THP-1 macrophages pretreated with the TLR3 inhibitor and si-TLR3 significantly reduced the phosphorylation of NF-κB p65 and the expression of NLRP3 (P < 0.01) (Fig 7B).

Subsequently, we verified the role of TLR3 in regulating the secretion of MIP-1β, IL-8, and CXCL1 in THP-1 macrophages after treatment with the TLR3 inhibitor and si-TLR3. The results revealed that THP-1 macrophages pretreated with the TLR3 inhibitor reduced the secretion of CXCL1 and MIP-1β in response to TV-EVs derived from both cell lines and the clinical strain (P < 0.001), and IL-8 secretion was significantly reduced in response to TV-EVs derived from the cell line (50143) (P < 0.001). There is no significant difference in IL-8 secretion for the clinical strain (NDMC5) upon treatment with the TLR3 inhibitor when compared to the EV-treated group (Fig 7C). Similarly, the intervention of si-TLR3 resulted in a significant reduction in TV-EV-induced CXCL1 and MIP-1β secretion in THP-1 macrophages (P < 0.05). Moreover, IL-8 secretion was also significantly decreased in response to TV-EVs isolated from both cell lines upon si-TLR3 intervention (p < 0.01), with no significant difference observed for the clinical strain (NDMC5) (Fig 7C). These results suggest that TV-EVs upregulate TLR3 to activate the NF-κB/NLRP3 signaling pathway, leading to enhanced secretion of pro-inflammatory cytokines by THP-1 macrophages.

## TV-EVs activate TLR3 to modulate the PI3K, NF-κB, p38 MAPK, and ERK pathways in Ect

We next verified whether TV-EVs induce TLR3 expression to modulate the downstream immune response in Ect. Obviously, stimulation of Ect with TV-EVs induced TLR3 expression from 4 to 10 hours (P < 0.05) (Fig 8A). We then explored whether TLR3 is involved in the activation of the PI3K, NF-κB, p38 MAPK, and ERK inflammatory pathways induced by

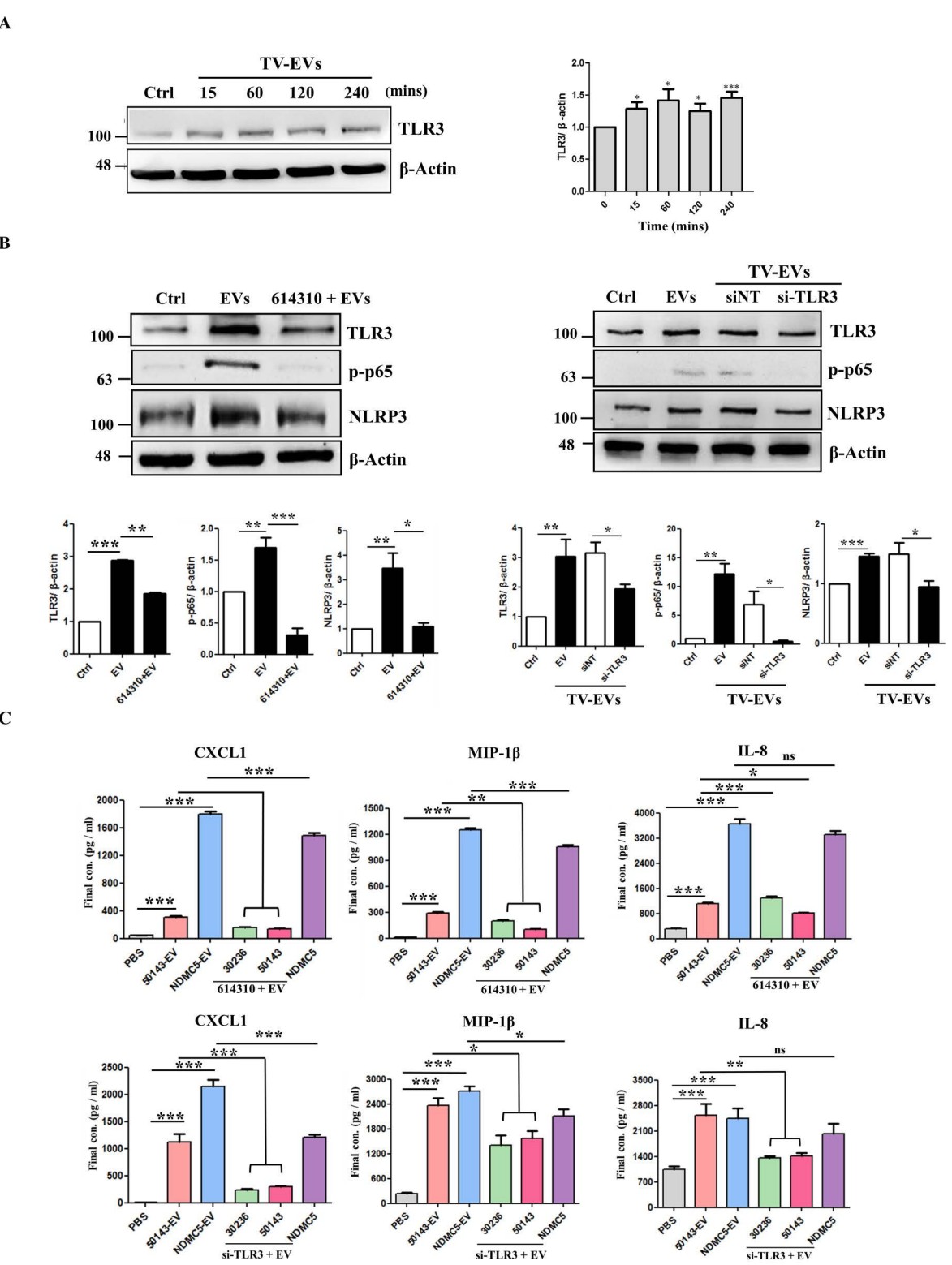

**Fig 7. TV-EVs activate the NF-κB/NLRP3 signaling pathway by TLR3 in THP-1 macrophages. (A)** THP-1 macrophages (5 × 10⁶ cells) were treated with the PBS control (Ctrl) or TV-EVs (ATCC 50143) for 15, 60, 120, and 240 minutes. Whole cell lysates were collected and analyzed by western blot

analysis using an anti-TLR3 antibody. Quantification of protein bands using Image J. **(B)** THP-1 macrophages were pretreated with the TLR3 inhibitor (614310) or siRNA targeting TLR3 (si-TLR3), and then co-cultured with TV-EVs for 15 minutes. The protein expression levels of TLR3, p-NF-κB, and NLRP3 in whole cell lysates of THP-1 macrophages were analyzed by western blot analysis using specific antibodies. Quantification of protein bands as mentioned above. **(C)** THP-1 macrophages ($2 \times 10^5$ cells/well) were pretreated the TLR3 inhibitor or si-TLR3, and then co-cultured with TV-EVs from different isolates for 6 hours. The culture supernatants were collected and analyzed for CXCL-1, MIP-1β and IL-8 by ELISA. Ect treated with the PBS control (Ctrl) and TV-EVs isolated from different strains (50143-EV and NDMC5-EV) were served as a negative and positive control, respectively. The protein expression of β-actin was used as the internal control for western blot analysis. The quantitative data are expressed as the means ± SEM from three independent experiments.*$p < 0.05$, ** $p < 0.01$, *** $p < 0.001$. ns: not significant.

TV-EVs in Ect. To address this, Ect was pretreated with the TLR3 inhibitor 614310, and the phosphorylation status of PI3K, NF-κB p65, p38 MAPK, and ERK was determined. It is noteworthy that the TV-EV-induced phosphorylation of PI3K and NF-κB p65 were significantly decreased (P < 0.05) in Ect pretreated with 614310, whereas the levels of p38 MAPK and ERK were increased (P < 0.05) (Fig 8B and 8C). Additionally, silencing TLR3 with si-TLR3 had a similar effect on the phosphorylation of these proteins (Fig 8B and 8C). Moreover, the secretion of inflammatory cytokines induced by TV-EVs was determined in Ect pretreated with the TLR3 inhibitor and si-TLR3. As shown in Fig 8D, IL-6 was markedly elevated (P < 0.05) after blocking TLR3 with the TLR3 inhibitor or si-TLR3, whereas IL-8 and CXCL1 exhibited apparent inhibition compared to the TV-EV-treated cells. These results revealed that TLR3 induces IL-8 and CXCL1 but inhibits IL-6 secretion in Ect. Overall, these findings suggest that TV-EV-induced TLR3 expression is involved in the inflammatory response by upregulating the PI3K and NF-κB pathways, while downregulating the p38 MAPK and ERK pathways in Ect.

## Proteomic analysis identify novel proteins regulating the TV-EV-induced inflammatory pathways in Ect

To further identify previously uncharacterized proteins or pathways that may be involved in regulating the immune response induced by TV-EVs, we conducted a proteomic analysis of Ect treated with TV-EVs. The differentially expressed proteins of Ect treated with TV-EVs purified from different strains (ATCC 30236, ATCC 50143, and NDMC5) for 2, 4, and 8 hours were analyzed and compared with the PBS-treated control. The enriched protein sets were identified by Gene Set Enrichment Analysis (GSEA) using Kyoto Encyclopedia of Genes and Genomes (KEGG) functional annotations (S2 Table). Pathway mapping revealed that TV-EVs upregulate pathways associated with 'Natural killer cell mediated cyto-toxicity, 'Cell adhesion molecules', and 'ECM receptor interaction', while concurrently downregulating pathways related to 'Vibrio cholerae infection', 'spliceosome', and 'Arrhythmogenic right ventricular cardiomyopathy' (Fig 9A and 9B). Addition-ally, many cancer-related pathways are upregulated in the Ect proteomes treated with EVs (S2 Table). In these enriched cancer-related protein sets, PIK3R1 (Phosphoinositide-3-Kinase Regulatory Subunit 1; PI3K p85) is the most upregulated protein. This finding is consistent with our previous results, reinforcing that PI3K signaling is activated in Ect in response to TV-EVs.

To validate the proteomic data, we assessed the expression levels of four proteins by western blot analysis, including MHC class I polypeptide-related sequence B (MICB), muscleblind like splicing regulator 2 (MBNL2), TRAF3 interacting protein 2 (TRAF3IP2), and kinesin light chain 4 (KLC4), which exhibit a 2-fold upregulation at least in two Ect proteomes treated with TV-EVs purified from three different strains (S3 Table). Remarkably, TV-EVs enhanced the protein expression levels of MICB, MBNL2, TRAF3IP2, and KLC4 in Ect after 2 hours of treatment, and this activation persisted for at least 8 hours (Fig 9C). MICB is one of the ligands for the natural killer group 2 member D protein (NKG2D) receptor expressed on the surface of NK cells and CD8 + T cells, and is induced in response to cellular stress [34]. Additionally, TRAF3IP2 has been shown to mediate high glucose-induced activation of NF-κB and downstream inflammation in endothelial cells [35]. Given that MICB and TRAF3IP2 have been demonstrated to regulate the host immune response [36–38] and are highly upregulated in Ect treated with TV-EVs, we investigated whether these two proteins are modulated by TLR3. The results showed that the expression levels of MICB and TRAF3IP2 were downregulated after si-TLR3 intervention, suggesting that TLR3 positively regulates MICB and TRAF3IP2 (Fig 9D). Next, we addressed whether MICB and TRAF3IP2 regulates

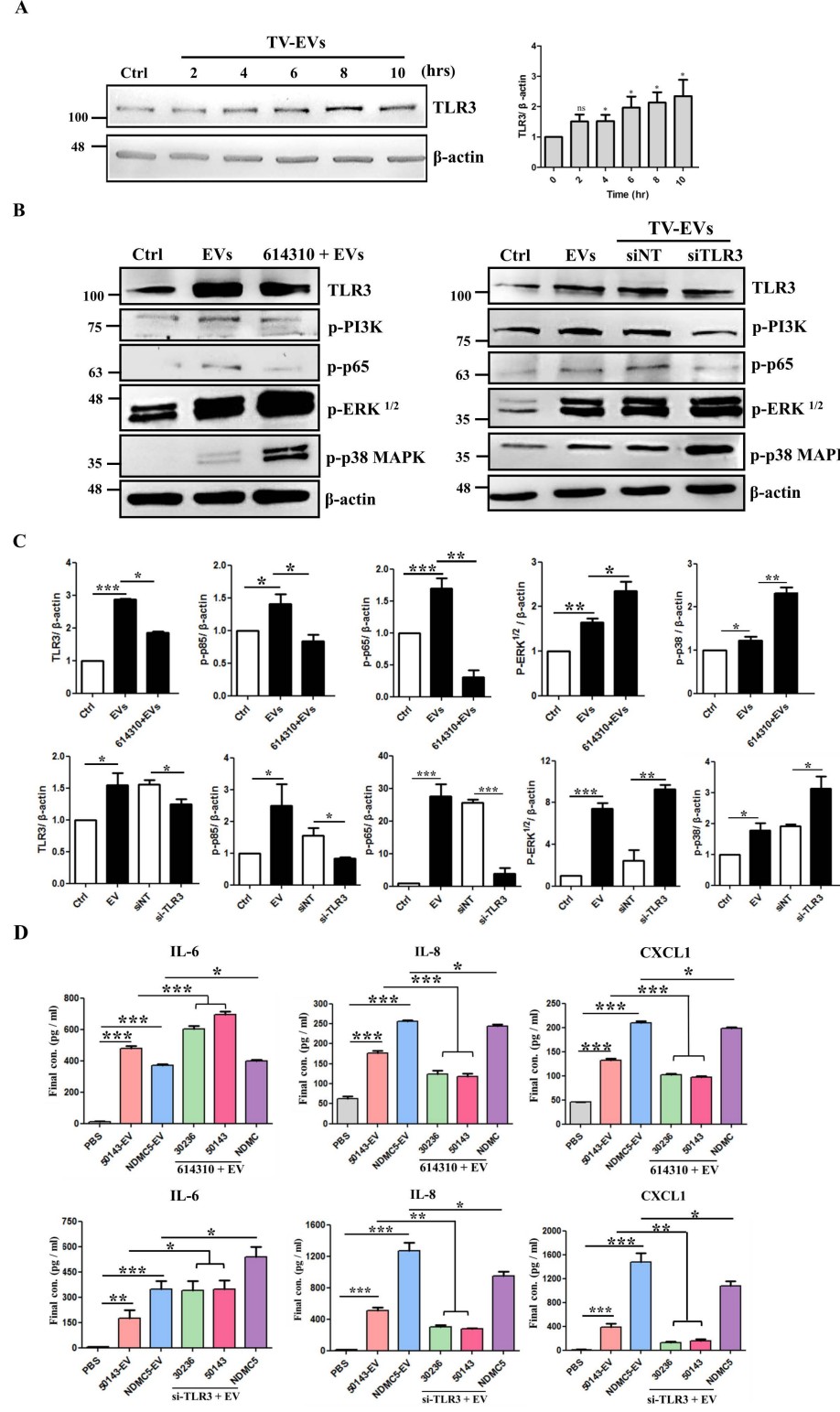

**Fig 8. TV-EVs upregulate TLR3 expression to modulate the PI3K, NF-κB, p38 MAPK, and ERK-mediated inflammation in Ect. (A)** Ect ($2 \times 10^6$ cells) were treated with the PBS control (Ctrl) or TV-EVs (ATCC 50143) for 2, 4, 6, 8, and 10 hours. Whole cell lysates were collected and analyzed by western blot using an anti-TLR3 antibody. Quantification of protein bands using Image J. **(B)** Ect were pretreated with the TLR3 inhibitor (614310)

or siRNA targeting TLR3 (si-TLR3) and then co-cultured with TV-EVs for 4 hours, and the protein expression levels of TLR3, p-PI3K, p-NF-κB, p-ERK, and p-p38 MAPK were analyzed using specific antibodies. **(C)** Quantification of protein bands from **(B)** using Image J. **(D)** Ect ($2 \times 10^5$ cells/well) were pretreated with the TLR3 inhibitor or si-TLR3, then co-cultured with TV-EVs from different isolates for 8 hours. The culture supernatants were collected and analyzed for IL-6, IL-8, and CXCL1 by ELISA. Ect treated with the PBS control (Ctrl) and TV-EVs isolated from different strains (50143-EV and NDMC5-EV) were served as a negative and positive control, respectively. The data are expressed as the means ± SEM from three independent experiments.*$p < 0.05$, **$p < 0.01$, ***$p < 0.001$.

the TV-EV-induced inflammatory pathways in Ect. The results indicated that the PI3K, NF-κB, and ERK signaling pathways were significantly inhibited in Ect after si-MICB intervention (Fig 9E). Additionally, Ect treated with si-TRAF3IP2 also significantly inhibited the PI3K and NF-κB pathways but remarkably activated the ERK pathway. Taken together, our data demonstrate that MICB and TRAF3IP2 are positively regulated by TLR3 and are involved in TV-EV-induced inflammatory cascades in Ect.

## Discussion

It has become clear that most cells can communicate within their immediate environment by releasing EVs. A previous study showed that exosomes derived from TV deliver regulatory molecules to the host and modulate the host immune response by inhibiting IL-8 secretion [21]. Additionally, exosomes derived from a highly adherent TV strain have been shown to enhance the adherence of a poorly adherent strain to epithelial cells. Another study in the murine model indicated that increased IL-10 production, combined with a decrease in IL-17 levels, resulted in reduced vulvar edema and inflammation in female mice pretreated with TV-EVs [25]. As the molecular mechanisms governing inflammation induced by TV-EVs in host cells remain poorly understood, our study unveils for the first time that TV-EVs activate the NLRP3 inflammasome and TLR3 expression to orchestrate the host immune response by modulating diverse pathways (Fig 10).

Previous studies have shown that TV-EVs are round-shaped vesicles with diameters ranging from 50 to 100 nm, which can be internalized by Ect and deliver their contents to host cells, consequently inducing the secretion of IL-6 and IL-8 [21]. A recent study reported that TV-derived EVs harbor a double-stranded RNA virus called TVV and induce the production of IL-8, IL-6, and IL-1β in human HaCaT cells [22]. Our results also indicated that TV secretes vesicles with a size of 100–120 nm, which can be taken up by both Ect and macrophages. Consistent with previous findings, our cytokine profiles showed that the secretion of IL-6, IL-8, and IL-1β was significantly elevated in host cells treated with EVs isolated from both cell lines and the clinical strain. Notably, we observed a significant release of CXCL1 in THP-1 macrophages and Ect stimulated with TV-EVs, especially those derived from the clinical strain (NDMC5). This observation suggests that TV-EVs isolated from different strains can induce varying intensities of host immune responses, potentially due to differences in their protein compositions. A previous study reported that exosome-like vesicles isolated from the TV GT-21 strain induces higher expression of IL-10, IL-6 and TNF-α in RAW264.7 macrophages [25], supporting that EVs from different TV strains may trigger distinct immune responses. CXCL1 is an important chemokine in the development of many inflammatory diseases, promoting neovascularization, inflammatory response, and tumor formation [39]. Recent research has demonstrated the involvement of CXCL1 in the progression of uterine cervical cancer (UCC), which is the fourth leading cause of cancer-related death in women [40]. Additionally, CXCL1 also plays a vital role in promoting the migration of UCC cells [41]. As trichomoniasis has been linked to an increased risk of UCC, we purpose that TV-EV-induced CXCL1 secretion may contribute to the progression of UCC, which requires further investigation.

Studies on TV-EVs have primarily focused on their roles in host cell attachment and immune responses. There are only a limited number of studies investigating the molecular mechanisms underlying host cell death and inflammation induced by TV-EVs. TV has been shown to induce mitochondria-dependent activation of caspases and phosphorylation of p38 MAPK, leading to apoptotic cell death in macrophages [42]. Additionally, TV has been reported to trigger caspase-1-mediated cleavage of gasdermin D, resulting in pyroptotic cell death in macrophages [11]. In our study, we demonstrated that TV-EVs activate the NLRP3 inflammasome and caspase-1 in THP-1 macrophages, leading to cytotoxic

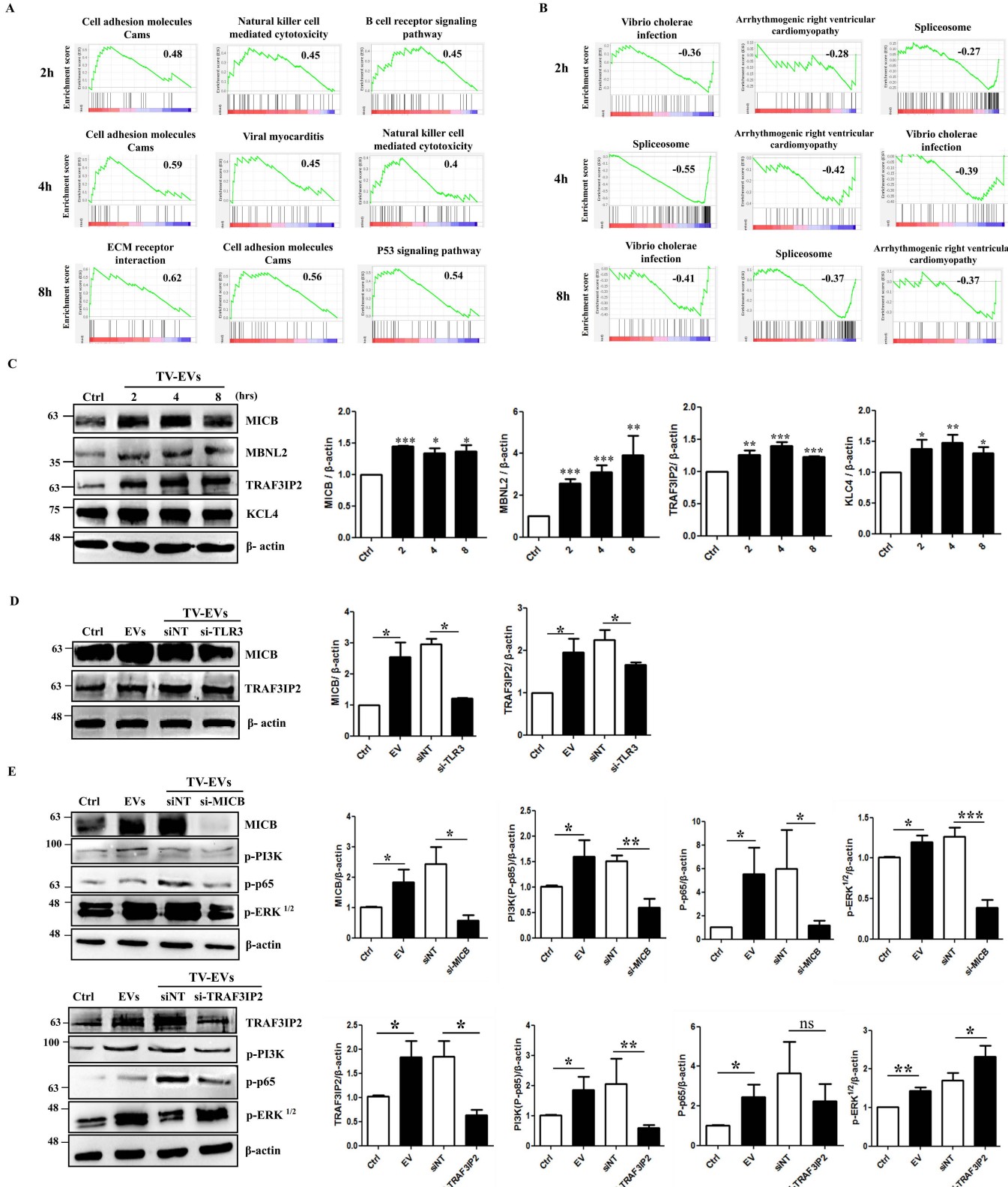

**Fig 9. Proteomic analysis identifies novel proteins involved in the TV-EV-induced immune response in Ect.** Enrichment analysis of differentially expressed proteins revealed enriched gene sets in the Ect proteome treated with TV-EVs purified from different strains for different time intervals (2,

PLOS Pathogens

4, and 8 hours). The enrichment plots showed the most **(A)** up-regulated or **(B)** down-regulated pathways in Ect treated with TV-EVs using the KEGG database. The enrichment score (ES) represents the degree to which the genes in the set are over-represented at either the top or bottom of the ranked list. The positive ES and negative ES indicate upregulation and downregulation of a specific gene set, respectively. **(C)** Validation of proteomic results by western blot analysis. Ect ($2 \times 10^6$ cells) were treated with the PBS control (Ctrl) or TV-EVs for 2, 4, and 8 hours, and whole cell lysates were collected and analyzed by western blot using antibodies against MICB, MBNL2, TRAF3IP2, and KLC4. **(D)** The role of TLR3 in regulating the expression levels of MICB and TRAF3IP2. Ect pretreated with the siRNA targeting TLR3 (si-TLR3) were co-cultured with TV-EVs for 4 hours, and the whole cell lysates were analyzed by western blot using specific antibodies. **(E)** The role of MICB and TRAF3IP2 in regulating the inflammatory pathways induced by TV-EVs. Ect pretreated with si-MICB and si-TRAF3IP2 were co-cultured with TV-EVs for 4 hours, and the whole cell lysates were analyzed using specific antibodies. Quantification of protein bands was performed using Image J. The protein expression of β-actin was used as the internal control for western blot analysis. The data are expressed as the means ± SEM from three independent experiments. $*p < 0.05$, $**p < 0.01$, $***p < 0.001$. ns: not significant.

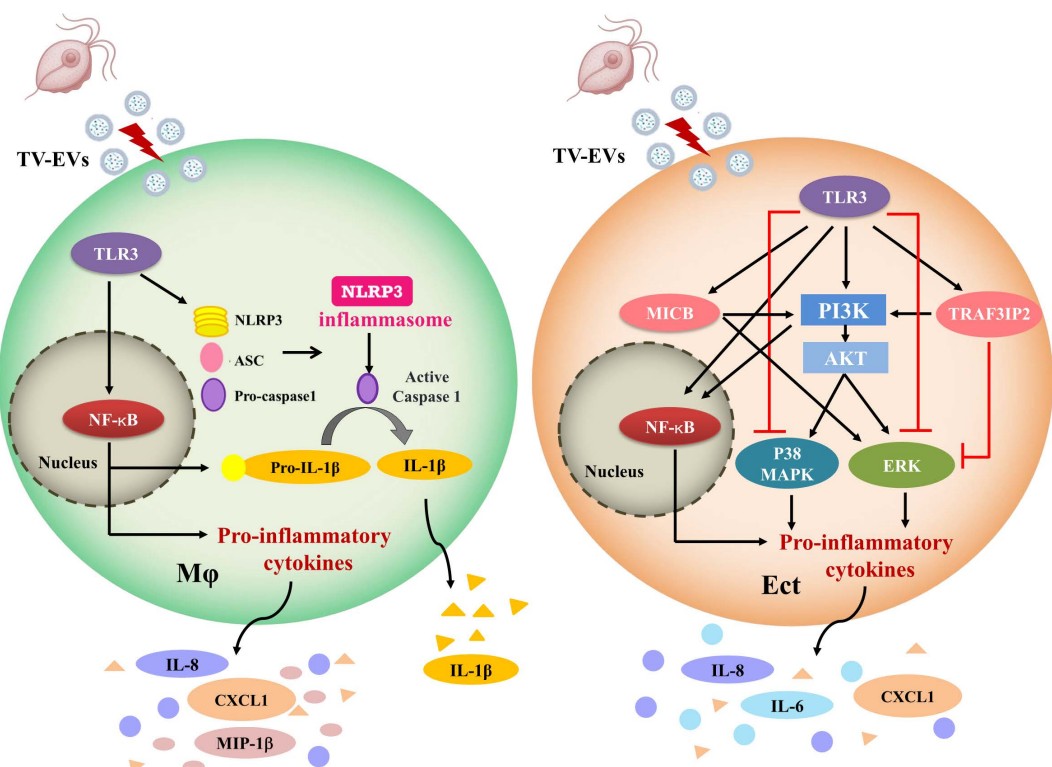

**Fig 10. Proposed model for TV-EV-induced inflammatory cascade in host cells.** In THP-1 macrophages, TV-EVs activate the TLR3-mediated NF-κB/NLRP3 pathway, inducing the secretion of IL-8, MIP-1β, CXCL1, and IL-1β. In Ect, TV-EVs activate PI3K signaling to positively regulate the NF-κB, p38 MAPK, and ERK pathways, inducing the secretion of IL-6, IL-8, and CXCL1. Specifically, TV-EV-induced TLR3 overexpression in Ect upregulates PI3K and NF-κB pathways while downregulating the p38 MAPK and ERK pathways. Furthermore, proteomic analysis identifies that TV-EVs induce overexpression of MICB and TRAF3IP2 in Ect, which are positively regulated by TLR3. MICB enhances PI3K-mediated activation of the NF-κB and ERK pathways, while TRAF3IP2 activates PI3K signaling but suppresses the ERK pathway.

effects in host cells upon prolonged exposure. These results suggest that TV-EVs may also induce pyroptosis in macrophages. Further investigations are warranted to elucidate the molecular mechanisms underlying host cell death in response to extended TV-EV exposure. TV has been demonstrated to induce IL-1β secretion by activating the NLRP3 inflammasome in THP-1 macrophages [11]. Similarly, we found that TV-EVs could promote IL-1β secretion. Activation of the NLRP3 inflammasome is typically associated with the secretion of both IL-1β and IL-18 [43]. Although IL-18 is also processed by caspase-1, IL-1β is more commonly assessed as a representative cytokine of NLRP3 inflammasome activation [44,45]. Hence, our analysis focused on IL-1β secretion in THP-1 macrophages treated with TV-EVs. Future studies

could investigate whether TV-EVs also promote IL-18 release, thereby providing a more comprehensive understanding of the TV-EV-induced immune response in host cells. Additionally, TV is capable of eliciting the production of IL-6, TNF-α, and IFN-γ in mouse macrophages by activating the MAPK and NF-κB pathways [16]. EVs released from *G. duodenalis* (GEVs) have also been shown to enhance the production of pro-inflammatory cytokines, including IL-6, TNF-α, and IL-1β, through the activation of NF-κB, p38 MAPK and ERK signaling pathways in mouse macrophages, while the AKT signaling pathway inhibits the secretion of these cytokines [46]. Furthermore, GEVs mediate the activation of the NLRP3 inflammasome to regulate the release of IL-1β in host cells [47]. Our results demonstrate that TV-EVs induce the secretion of MIP-1β, IL-8, CXCL1, and IL-1β via the NF-κB/NLRP3 pathways in THP-1 macrophages. In Ect, TV-EV-induced IL6, IL8 and CXCL1 secretion is enhanced through the activation of NF-κB, p38 MAPK, and ERK pathways via PI3K signaling. In addition to the contact-dependent mechanism of TV for NLRP3 inflammasome activation in THP-1 macrophages [11], our work has identified a contact-independent mechanism of TV-EVs for NLRP3 inflammasome activation. We observed that NF-κB is a key orchestrator of the host immune response during TV infection, as TV-EV treatment activates NF-κB signaling in both THP-1 macrophages and Ect. Additionally, inhibition of NF-κB signaling dramatically reduced the secretion of TV-EV-induced pro-inflammatory cytokines. Hence, targeting the NF-κB signaling pathway may be a pivotal focus for drug discovery and development in the context of TV infection or TV-associated STIs and UCC.

TLRs play crucial roles in mediating immune responses and inflammatory pathways [48–50]. In addition to NF-κB, which serves as a primary transcription factor activated by TLR signaling, TLRs also trigger the p38 MAPK and ERK signaling pathways [51,52]. Previous studies have reported that *Leishmania* infection upregulates the expression of TLR1, TLR2, TLR3, and TLR4, and parasite-derived EVs possess immunomodulatory properties capable of inducing cytokine secretion via TLR2 and TLR4 [53,54]. In HeLa cells, TV stimulation upregulates the mRNA expression of TLR2, TLR4, and TLR9 through the ERK, p38 MAPK, and NF-κB pathways [15]. Additionally, TV induces the production of IL-6, TNF-α, and IFN-γ in mouse macrophages through the TLR2-mediated MAPK and NF-κB pathways [16]. Our results provide the first evidence that TV-EVs induce TLR3 overexpression in host cells, subsequently activating the NF-κB/NLRP3 signaling pathway in THP-1 macrophages, as well as the PI3K and NF-κB pathways in Ect. TLR3 is a member of the TLR family, recognizing double-stranded RNA (dsRNA) and regulating the transcription of pro-inflammatory cytokines to initiate an antiviral response [55,56]. A previous study indicated that TLR3-dependent pro-inflammatory activation is a common trait for TVV-infected TV isolates. Additionally, recent research demonstrated that TV-EVs contain viral RNA, eliciting a host immune response [22]. Based on these findings, it is likely that TVV-positive EVs release viral dsRNA into host cells, where it is recognized by TLR3, subsequently inducing a pro-inflammatory response. However, in the present study, we observed that EVs purified from a TVV-negative TV (ATCC 50143) [23] also induce TLR3 overexpression and TLR3-dependent NF-κB signaling in THP-1 and Ect. Hence, we hypothesize that components other than TVV viral dsRNA within TV-EVs or on the surface of TV-EVs can bind to TLR3 and induce downstream inflammatory signaling.

We observed that NDMC5 EVs did not significantly alter IL-8 secretion in THP-1 macrophages pretreated with a TLR3 inhibitor or si-TLR3, suggesting that TLR3 may not be involved in the regulation of NDMC5 EV-induced IL-8 secretion. This finding further supports the idea that differences in the protein composition of EVs may activate distinct signaling pathways, leading to varied regulation of inflammatory cytokine secretion in host cells. It is worth noting that TV-EV-induced IL-6 secretion was further increased in Ect pretreated with a TLR3 inhibitor or si-TLR3 intervention, while IL-8 and CXCL1 were significantly reduced. Additionally, phosphorylation of p38 MAPK and ERK was significantly induced in Ect pretreated with a TLR3 inhibitor or si-TLR3, while the PI3K and NF-κB signaling were inhibited under the same conditions. These results suggest that TV-EVs induce TLR3 overexpression, which activates the PI3K and NF-κB pathways, subsequently leading to the production of inflammatory cytokines, such as IL-8 and CXCL1. On the other hand, it is possible that TLR3 overexpression induced by TV-EVs may partially reduce IL-6 secretion by inhibiting the p38 MAPK and ERK pathways. Thus, TLR3 positively regulates specific inflammatory pathways to mediate TV-EV-induced inflammation, while concurrently exerting a negative regulatory role in other TV-EV-induced inflammatory pathways.

Mass spectrometry techniques facilitate the analysis of the protein composition of EVs, providing valuable insights into their roles in various diseases [57]. We used proteomic analysis to identify specific proteins involved in the TV-EV-induced inflammatory response in host cells. Our findings unveiled that TV-EVs trigger overexpression of MICB and TRAF3IP2 in Ect. MICB serves as a ligand for the NKG2D type II receptors present on NK cells and CD8 + T cells, mediating antitumor response and immune surveillance [58]. It is known that NKG2D ligands are expressed on the cell surface of stressed or damaged cells, facilitating recognition and killing by NKG2D-expressing cytotoxic cells [59]. Additionally, MICA/B expression is elevated in cervical cancer compared to low-grade cervical intraepithelial neoplasia and serves as an independent predictor of good prognosis in cervical cancer [60]. Further investigation is needed to determine whether MICB-overexpressing Ect can be targeted by NK cells. TRAF3IP2 has been shown to serve as a pro-inflammatory cytoplasmic adaptor protein, activating the NF-κB, MAPK and JNK signaling pathways [61]. Additionally, TRAF3IP2 also contributes to stabilizing the mRNA of CXCL1, which is involved in the angiogenesis, inflammatory response, and tumor formation. We have demonstrated that TV-EV-induced TLR3 expression positively regulates MICB, which further activates the inflammatory pathways in Ect, including the PI3K, NF-κB, and ERK pathways. Additionally, TLR3-dependent TRAF3IP2 expression is also involved in activating the PI3K and NF-κB pathways but suppresses the ERK pathway. It remains to be determined whether MICB and TRAF3IP2 regulate other signaling pathways or specific cytokine secretion in Ect stimulated with TV-EVs.

Our study demonstrates that TV-EVs induce the secretion of pro-inflammatory cytokines CXCL1, MIP-1β, and IL8 through the NF-κB/NLRP3 signaling, which is regulated by TLR3 in THP-1 macrophages. Additionally, TV-EVs elicit the release of IL6, IL8, and CXCL1 through TLR3-mediated activation of the PI3K and NF-κB pathways in Ect. Proteomic analysis provides the first evidence of the roles played by MICB and TRAF3IP2 in TV-EV-induced immune responses in Ect. Our study sheds light on previously uncharacterized mechanisms of inflammation in host cells induced by TV-EVs and identifies novel proteins modulating the immune response. These findings significantly advance our understanding of the crosstalk between TV-EVs and the host.

## Materials and methods

### Ethics statement

This study was approved by the Institutional Review Board of Tri-Service General Hospital, National Defense Medical Center (TSGHIRB No.: A202105090), with permission to use residual samples. The formal written consent was obtained from the participants.

### Cell culture

TV cell lines (ATCC 30236 and ATCC 50143) were cultured in YI-S medium, pH 5.8, containing 10% heat-inactivated horse serum and 1% glucose at 37 °C. The TV clinical strain (NDMC5) was isolated from urine specimens, initially screened by medical technologists in the Department of Clinical Pathology at Tri-Service General Hospital. The clinical isolate was cultured in the same YIS medium in the presence of Antibiotic-Antimycotic Solution (100X) (Sigma-Aldrich, USA). Growth of the parasites was monitored by using trypan blue exclusion hemocytometer counts. The human ecto-cervical epithelial cell line Ect1/E6E7 (ATCC CRL-2614) was cultured in keratinocyte serum-free medium (KSFM) (Gibco, Thermo Fisher Scientific, Inc., Waltham, MA, USA) supplemented with 0.1 ng/mL human recombinant epidermal growth factor, 0.05 mg/mL bovine pituitary extract, and 0.4 mM calcium chloride [62]. The human monocyte cell line THP-1 (ATCC TIB-202) was cultured in Roswell Park Memorial Institute medium (RPMI 1640) supplemented with 10% fetal bovine serum (FBS) and incubated in 5% $CO_2$ at 37°C. THP-1 cells are differentiated in vitro into macrophages by the administration of 50 ng/ml phorbol myristate acetat (PMA).

## Isolation of TV-EVs

TV ($2 \times 10^6$ parasites/ml) were washed and resuspended in YI-S medium without serum for 8 hours. The parasites were removed and the supernatants were collected by centrifugation at 3,000 rpm for 10 minutes. The supernatants were filtered through a 0.22 µm sterile disposable bottle (Thermo Scientific, USA) and concentrated by centrifugal filter units (Amicon Ultra-15, Merck Millipore, USA). The total exosome isolation reagent (Life Technologies, Carlsbad, CA, USA) was added to the supernatants and incubated at 4 °C overnight, with a supernatant-to-reagent ratio of 1:0.33. After incubation, the supernatants were centrifuged at $10,000 \times g$ for 1 hour. Pellets were resuspended in 50 µl PBS and stored at -80 °C. To analyze the concentration and particle size of EVs, nanoparticle tracking analysis (NTA) was performed using the NanoSight NS300 instrument (NTA, NanoSight Ltd. Amesbury, UK).

## Transmission electron microscopy

EVs purified from different TV cell lines and the clinical isolate were added to charged carbon-coated grids after being stained with 1% uranyl acetate. TV-EVs treated with uranyl acetate were examined using a transmission electron microscope (Hitachi HT-7700, Hitachi. Co. Ltd., Japan).

## Internalization of TV-EVs with THP-1 and Ect

Purified TV-EVs were labeled using the PKH67 Green Fluorescent Cell Linker Kit (Sigma-Aldrich, USA) according to the manufacturer's manual with modifications. Briefly, EVs were mixed with an equal volume of diluent C and PKH67 dye in 150 µl of PBS at room temperature for 3 minutes in the dark. The reaction was terminated by adding 0.1% bovine serum albumin (BSA). The PKH67-labeled EVs were centrifuged at 14,000 rpm for 1 hour at 4 °C, and the pellets were collected and washed with PBS at 14,000 rpm for 30 minutes at 4 °C. The PKH67-labeled EVs were obtained and resuspended in PBS. THP-1 and Ect ($3 \times 10^4$ cells) were seeded on an 18 mm × 18 mm coverslip at 37°C for 24 hours in a 5% $CO_2$ incubator. After incubation, cells were coincubated with PKH67-labeled EVs (30µg/ml) for different time intervals (30 minutes, 1 hour, 2 hours, 3 hours, and 6 hours). The cells were washed with PBS three times to remove any uninternalized EVs. The cells were then stained with the plasma membrane staining reagents (CellMask Deep Red, Invitrogen, USA) for 10 minutes, fixed with 4% formaldehyde for 30 minutes, and stained with DAPI. The internalization of TV-EVs by THP-1 macrophages and Ect was visualized with confocal microscopy (LSM 980, Zeiss, Germany) and TEM (Hitachi HT-7700, Hitachi. Co. Ltd., Japan).

## Cell viability assay

Cell viability was assessed by a colorimetric assay using the CCK-8 Kit (Sigma-Aldrich, USA). THP-1 macrophages or Ect were seeded in 24-well plates ($2 \times 10^5$ cells/well) overnight. TV-EVs (30µg/ml) were added to the wells and co-cultured with the cells in serum-free medium for various time intervals. Finally, 20 µl of the CCK-8 reagent was added to each well, and absorbance was determined at 450 nm using a microplate reader after incubation for 2 hours at 37 °C.

## Multiplex immunoassay

To screen the secretion of multiple cytokines from host cells treated with TV-EVs, Ect and THP-1 macrophages ($2 \times 10^5$ cells/well) were co-cultured with TV-EVs (30µg/ml) for 6 and 8 hours in serum-free medium, respectively. The samples were centrifuged at 3,000 rpm for 10 minutes to remove the cells. The multiplex immunoassay (Bio-Rad, USA) was performed to measure the production of 48 common human cytokines in the culture supernatants.

## Enzyme-linked immunosorbent assay (ELISA)

To confirm the production of inflammatory cytokines (IL-1β, IL-6, IL-8, MIP-1β, and CXCL1) from host cells stimulated with TV-EVs, THP-1 macrophages or Ect were seeded at a density of $2 \times 10^5$ cells/well in 24-well plates and co-incubated with

TV-EVs for 2, 4, 6, 8, and 10 hours. After incubation, the supernatant was collected and stored at -20 °C. The levels of IL-1β, IL-6, IL-8, MIP-1β, and CXCL1 were measured using ELISA kits (R&D System, USA) and quantified by detecting absorbance at 450 nm with an ELISA reader.

### RNA extraction and cDNA synthesis

Total RNA was isolated from Ect using the SV Total RNA Isolation System (Promega, UK) according to the manufacturer's protocol. cDNA was generated by reverse transcription of RNA extracted from TV-infected and non-infected (control) Ect using the First-Strand Synthesis System (Invitrogen, USA). Firstly, RNA was mixed with dNTP, Oligo (dT), and DEPC-treated water, and incubated at 65°C for 5 minutes. 10X RT Buffer, 0.1 M DTT, RNaseOUT, and SuperScript II RT were added, and the reaction was incubated at 50°C for 50 minutes. The reaction was terminated at 85°C for 5 minutes. Finally, RNase H was added, and the reaction was incubated for 20 minutes at 37°C. The cDNA products were stored at -20 °C until use.

### Reverse transcription-PCR

Reverse transcription PCR (RT-PCR) was performed using SuperScript One-Step RT-PCR with GoTaq Colorless Master Mix (Promega, USA) to investigate the gene expression of TLRs. All of the cDNA was synthesized from 1 µg of total RNA extracted from Ect. Next, the RT-PCR products mixed with the DNA staining reagent (Novel Juice) (GeneDireX, USA) were separated using a 1.7% agarose gel. Primer pairs used in this study are listed in S1 Table.

### Immunofluorescence assay

Ect or THP-1 macrophages ($4 \times 10^4$ cells) were cultured on glass coverslips overnight and then treated with TV-EVs (30µg/ml). Next, cells were washed with PBS, fixed with 4% formaldehyde and then incubated for 1 hour with one of the following antibodies at a concentration of 1:200 diluted in blocking buffer: rabbit anti-human NF-κB p65 (ABclonal, Germany), rabbit anti-human NLRP3 (Cell Signaling Technology Inc., Danvers, MA, USA), or rabbit anti-human ASC (Abcam, UK). Alexa Fluor 488 goat-rabbit IgG was used as the secondary antibody at a dilution of 1:500. Nuclei were counterstained with DAPI (Sigma-Aldrich, USA), followed by examination under a confocal microscope (LSM 980, Zeiss, Germany).

### Inhibitor treatments

To inhibit the TLR3, PI3K, p38 MAPK, ERK, and NF-κB pathways, Ect ($2 \times 10^6$ cells) were pretreated with 100 µM TLR3 inhibitor 614310 (Sigma-Aldrich, USA), 10 µM PI3K inhibitor wortmannin (Sigma-Aldrich, USA), 10 µM p38 MAPK inhibitor SB203580 (InvivoGen, USA), 50µM ERK inhibitor PD98059 (Sigma-Aldrich, USA) or 100 µM NF-κB inhibitor BAY 11–7082 (Sigma-Aldrich, USA), respectively, for 1 hour in serum-free medium.

### RNA interference

Small interfering RNA (siRNA) (Horizon Discovery, UK) was prepared for the target genes TLR3, MICB, and TRAF3IP2, and a non-targeting siRNA was served as the negative control (siNT) for non-specific effects. The 20 µM siRNA stock solution was diluted with serum-free KSFM to a working concentration of 5 µM. Thereafter, 5 µl of the siRNA dilution was added to 95 µl of serum-free KSFM (25 nM concentration), mixed, and incubated for 5 minutes. DharmaFECT reagent (Horizon Discovery, UK) was diluted 20 times with serum-free KSFM and incubated for 5 minutes. Next, both dilutions were mixed and incubated for 20 minutes. The working dilution was added to Ect ($1 \times 10^6$ cells/well) or THP-1 macrophages ($2 \times 10^6$ cells/well) and incubated at 37 °C with 5% $CO_2$. The working dilution was replaced with ordinary KSFM after incubation for 8 hours. The Ect were maintained until the transfection process was completed. Finally, we performed the subsequent experiments after transfection with siRNA for 72 hours.

## LC–MS/MS analysis for protein identification

Protein solutions were diluted in 200 mM TEABC, reduced with 5 mM TCEP (Sigma-Aldrich, USA) at 60°C for 45 minutes, then cysteine-blocked with 10 mM MMTS (Sigma-Aldrich, USA) at 25°C for 30 minutes. Sequencing-grade modified porcine trypsin (Promega, USA) was used for protein digestion occurred at 37°C for 16 hours. Peptides were labeled with TMT reagent for 1 hour at room temperature, pooled, desalted by homemade C18-microcolumn (SOURCE 15RPC, GE Healthcare, USA), then dried by vacuum centrifugation, and stored at −20 °C until use. The dried peptide mixtures were reconstituted and loaded onto a homemade column (Luna SCX 5μm, 0.38 × 400 mm) at flow rate of 2 μl/minute for 60 minutes. The peptides were then fractionated to 70 fractions by eluting with 0–100% HPLC buffer B (1 M ammonium nitrate/25% acetonitrile/0.1% formic acid) using on-line 2D-HPLC (Dionex Ultimate 3000, Thermo Fisher, USA). Each SCX fraction was diluted in-line prior to trap onto a reverse-phase column (Zorbax 300SB-C18, 0.3 × 5 mm; Agilent Technologies, USA). The peptides were then separated on a column (Waters BEH 1.7 μm, 100 μm I.D. × 10 cm with a 15 μm tip) using a multi-step gradient of HPLC buffer C (99.9% acetonitrile/0.1% formic acid) for 62 minutes with a flow rate of 0.3 μl/minute. The LC apparatus was coupled with a 2D linear ion trap mass spectrometer (LTQ-Orbitrap ELITE; Thermo Fisher, USA) operated using Xcalibur 2.2 software (Thermo Fisher, USA). The full-scan MS was performed in the Orbitrap over a range of 400–2,000 Da and a resolution of 120,000 at m/z 400. For proteome analysis, the 10 data-dependent MS/MS scan events were followed by one MS scan for the 10 most abundant precursor ions in the preview MS scan. The m/z values selected for MS/MS were dynamically excluded for 60 seconds with a relative mass window of 1 Da. The electrospray voltage was set to 2.0 kV, and the temperature of the capillary was set to 200°C. MS and MS2 automatic gain control were set 1,000 ms (full scan), and 300 ms (MS2 for HCD), or $3 \times 10^6$ ions (full scan), and $3 \times 10^4$ ions (MS2 for HCD) for maximum accumulated time or ions, respectively.

## Protein identification and quantification

MS data were processed with Proteome Discoverer software (v2.3, Thermo Fisher Scientific) for protein identification and TMT quantification. MS/MS spectra were searched against Swissprot using Mascot (v2.5, Matrix Science). Peptide identification allowed 10 ppm mass tolerance for intact peptides and 0.05 Da for HCD fragment ions, with two missed cleavages permitted. Variable modifications included oxidation (M), acetyl (protein N-terminal), and TMT10plex (peptide N-terminal, K), with methylthio (C) as static modification. Peptide-spectrum match (PSM) were filtered for high confidence (rank 1) to ensure an FDR below 0.01. Proteins with single peptide hits were excluded. Quantitative protein data were exported using centroid integration with 20 ppm tolerance.

## Bioinformatics analysis of the proteomic data

Differential protein expression analysis in the Ect proteomes treated with TV-EVs (ATCC 30236, ATCC 50143, and NDMC5) for 2, 4, and 8 hours, compared with the PBS-treated control, was performed using Gene Set Enrichment Analysis (GSEA). Kyoto Encyclopedia of Genes and Genomes (KEGG) [63] was incorporated into the GSEA software as reference databases to identify the enriched protein sets of all identified proteins.

## Nuclear extraction

Nuclear extraction was performed using a nuclear extraction kit (Abcam, UK). THP-1 macrophages ($5 \times 10^6$ cells/ml) were co-cultured with TV-EVs for 4 hours in serum-free medium. After incubation, the cells were washed three times with PBS and collected by centrifugation at 1,000 rpm for 5 minutes. The cells were resuspended in 500 μl of 1X Pre-Extraction Buffer and placed on ice for 10 minutes, followed by centrifugation at 12,000 rpm for 2 minutes at 4°C. The supernatant was collected as the cytoplasmic fraction. The pellet was then resuspended in 50 μl of Extraction Buffer and incubated on

ice for 15 minutes, with vortexing for 5 seconds every 3 minutes. Finally, the mixture was centrifuged at 14,000 rpm for 10 minutes at 4°C, and the supernatant was collected as the nuclear fraction.

## Immunoblot analysis

THP-1 macrophages ($5 \times 10^6$ cells) or Ect ($2 \times 10^6$ cells) were seeded in a 6 cm cell culture dish overnight. The cells were co-incubated with TV-EVs (30 μg/ml) for different time intervals, followed by lysis with RIPA buffer (Bio-Future, Taiwan) containing protease inhibitor cocktail (Bio-Future, Taiwan). Protein samples were separated by sodium dodecyl sulfate polyacrylamide gel electrophoresis (SDS-PAGE). The following primary antibodies were used for immunoblot analysis: antibodies against TLR3 (1:1000; ABclonal, Germany), IL-1β (1:1000; R&D systems, USA), NLRP3, ASC, Caspase-1, phosphor-NF-κB p65, phosphor-p44/42 MAPK (p-ERK$^{1/2}$), phosphor-p38 MAPK, phosphor-PI3K(p85), phosphor-AKT, MICB, and MBNL2 (1:1000; Cell Signaling Technology Inc., Danvers, MA, USA), NF-κB p65 (1:1000; ABclonal, Germany), TRAF3IP2 and KLC4 (1:1000; Invitrogen, USA), and β-actin (1:2000; Cell Signaling Technology Inc., Danvers, MA, USA) at 4 °C overnight. Goat anti-rabbit horseradish peroxidase-conjugated IgG (1:5000; Cell Signaling Technology, Beverly, MA, USA) was used as the secondary antibody. After washing three times with tris-buffered saline with 0.1% tween (TBST, Bio-Future, Taiwan), the membrane was soaked in chemiluminescent HRP substrate (Merck Millipore, USA) for signal development. Quantification of protein bands from the immunoblot data was performed using Image J.

## Animal experiments

The animal experiments for TV-EVs infection were based on previous mouse models of *T. vaginalis* infection [11,25], with modifications. Seven-week-old female BALB/c mice were purchased from BioLASCO (Taipei, Taiwan). The mice were pretreated with daily intraperitoneal (i.p.) injections of 100 μl Dexamethasone Sodium Phosphate (2 mg/ml) (Decan, Yung Shin, Taiwan) for 5 days prior to infection. The day before infection, the mice were also injected i.p. with 100 μl β-estradiol (5 mg/mL) resuspended in sesame oil (Sigma-Aldrich, USA). TV-EVs isolated from $2 \times 10^7$ TV (ATCC 50143 and NDMC5) were resuspended in PBS, and a total volume of 15–20 μl of TV-EVs was used to infect the mice vaginally for 2 consecutive days. Vaginal lavages were collected every three days until the twelfth day by washing the vaginal lumen four times with 50 μl PBS using gel-loading pipette tips.

## Statistical analysis

Quantitative data were expressed as mean ± SEM of three independent experiments. Comparisons of two groups were performed using Student's *t*-test (two-tailed) for paired samples. Statistical analysis was performed using GraphPad Prism 5 software (GraphPad Inc., La Jolla, CA, USA). Values of $p < 0.05$ was considered statistically significant (*) and $p < 0.01$ (**) or $p < 0.001$ (***) to be very statistically significant.

## Supporting information

**S1 Fig. Effects of TV-EVs on the viability of THP-1 macrophages and Ect.** The morphological changes in (A) THP-1 macrophages ($2 \times 10^5$ cells/ml) and (B) Ect ($2 \times 10^5$ cells/ml) treated with TV-EVs (30 μg/ml) isolated from the cell line (ATCC 30236) and the clinical strain (NDMC5) for different time intervals were observed. (C) THP-1 macrophages ($2 \times 10^5$ cells/ml) or (D) Ect ($2 \times 10^5$ cells/ml) were co-cultured with TV-EVs (30 μg/ml) purified from different strains for different time intervals, and the cell viability was determined using the CCK-8 assay. Scale bar: 1μm. **$p < 0.01$, ***$p < 0.001$. (TIFF)

**S2 Fig. Nuclear translocation of NF-κB in THP-1 macrophages treated with TV-EVs.** THP-1 macrophages ($5 \times 10^6$ cells/ml) were treated with the PBS control (Ctrl) or TV-EVs (ATCC 50143) for 4 hours and the expresssion of NF-κB p65

was analysed in the cytoplasmic (Cy) and nuclear (Nu) fractions using western blot. Coomassie blue staining served as the loading control, and β-actin was used as a cytoplasmic marker to confirm fraction purity.
(TIFF)

**S3 Fig. The gene expression of TLR1–10 in Ect treated with TV and TV-EVs for different time intervals.** (A) The gene expression of TLR1–10 in Ect was screened by RT-PCR using specific primers. (B) The gene expression of TLRs in Ect stimulated with TV for different time intervals was measured by RT-PCR analysis. (C) The protein expression of TLR1, TLR3 and TLR5 in Ect treated with TV-EVs for different time intervals was detected by western blot.
(TIF)

**S1 Table. Primers used in this study.**
(XLSX)

**S2 Table. KEGG pathway enrichment analysis of differentially expressed proteins in Ect treated with TV-EVs purified from various strains for different time intervals.**
(XLSX)

**S3 Table. Differentially expressed proteins with a 2-fold upregulation were identified in at least two TV-EV-treated Ect proteomes after 2, 4, and 8 hours compared to PBS-treated control (Ctrl).**
(XLSX)

**S1 Data. Raw data for western blots.** This file includes raw image data of Figs 1A, 1B, 4A, 5A, 5D, 6A, 6D, 7A, 7B, 8A, 8B, 9C, 9D and 9E.
(PDF)

**S2 Data. Statistical Analysis.** This file includes raw data of statistical analysis in Figs 3A, 3B, 3C, 3D, 3E, 3F, 3G, 3H, 3J, 4B, 4C, 5B, 5D, 5E, 6B, 6D, 6F, 7A, 7B, 7C, 8A, 8C, 8D, 9C, 9D, 9E, S1C and S1D.
(XLSX)

## Acknowledgments

We thank the Proteomics Core Lab and Bioinformatics Core Laboratory, Molecular Medicine Research Center, Chang Gung University, Taiwan for technical support.

## Author contributions

**Conceptualization:** Ching-Yun Huang, Kuo-Yang Huang.

**Data curation:** Ching-Yun Huang, Shu-Fang Chiu, Lichieh Julie Chu, Po-Jung Huang, Kuo-Yang Huang.

**Formal analysis:** Ching-Yun Huang, Shu-Fang Chiu, Yuan-Ming Yeh, Po-Jung Huang, Ching-Chun Liu, Kuo-Yang Huang.

**Funding acquisition:** Shu-Fang Chiu, Kuo-Yang Huang.

**Investigation:** Ching-Yun Huang, Lichieh Julie Chu, Kuo-Yang Huang.

**Methodology:** Ching-Yun Huang, Lichieh Julie Chu, Yuan-Ming Yeh, Po-Jung Huang, Kuo-Lun Lan, Yu-Ling Tsai, Chien-Fu F Chen.

**Project administration:** Ching-Yun Huang, Kuo-Yang Huang.

**Resources:** Kuo-Yang Huang.

**Software:** Yuan-Ming Yeh.

**Supervision:** Kuo-Yang Huang.

**Validation:** Ching-Yun Huang, Shu-Fang Chiu, Kuo-Yang Huang.

**Visualization:** Ching-Yun Huang.

**Writing – original draft:** Ching-Yun Huang, Lichieh Julie Chu, Kuo-Yang Huang.

**Writing – review & editing:** Ching-Yun Huang, Wei-Ning Lin, Kuo-Yang Huang.

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
