## [Decision Letter · Decision Letter 0]

Dear Prof. Huang,

Thank you very much for submitting your manuscript "Trichomonas vaginalis extracellular vesicles activate the NLRP3 inflammasome and TLR3-mediated inflammatory cascades in host cells" for consideration at PLOS Pathogens. As with all papers reviewed by the journal, your manuscript was reviewed by members of the editorial board and by several independent reviewers. In light of the reviews (below this email), we would like to invite the resubmission of a significantly-revised version that takes into account the reviewers' comments.

Please address all of the Reviewer's comments

We cannot make any decision about publication until we have seen the revised manuscript and your response to the reviewers' comments. Your revised manuscript is also likely to be sent to reviewers for further evaluation.

Sincerely,

Keke C. Fairfax, PhD

Academic Editor

PLOS Pathogens

James Collins III

Section Editor

PLOS Pathogens

Michael Malim

Editor-in-Chief

PLOS Pathogens

orcid.org/0000-0002-7699-2064

Please address all of the Reviewer's comments

Reviewer's Responses to Questions

**Part I - Summary**

Reviewer #1: General comments: Three Trichomonas vaginalis strains were used, one of which was isolated from a patient. (2019) revealed that Trichomonas vaginalis induces the NLRP3 inflammasome in THP-1, and it was also reported that Tv-EV is internalized into ectocervical cells. TLR2, TLR3, TLR4 and TLR5 have been reported to be involved in the secretion of proinflammatory cytokines and chemokines from cervicovaginal mucosa. There is also a paper reporting proteomic analysis of EVs conducted by the Petricia group. However, in this paper, a total of three TVs were used, including Trichomonas vaginalis isolated from a patient, and this paper appears to be the first to reveal that TLR3 is involved in the induction of NLRP3 inflammasome in THP-1 cells by Tv. In addition, inflammasome-related experiments were conducted well, and additional proteomic data was used to identify key substances using western blot.

Points to improve: In this experiment, THP-1 cells and ectocervical cells were treated with Ev isolated from three strains of Trichomoniasis vaginalis, and the downstream signals of the inflammatory response within the cells were examined. However, the experiments on THP-1 cells and ectocervical cells were not conducted in parallel and in pairs, and in some experiments, only one cell was used, which overall feels disorganized. According to the authors' results, the micrographs appear to have been made using a general fluorescence microscope, and the resolution is low, so they do not accurately show the distribution and location of Ev, p65 NF-kB, etc. within the cell. And, some western results show that the expression of p65Nf-kB is too weak.

Reviewer #2: The authors reported an interesting array of results aiming to understand molecular mechanisms of Tv-derived EV mediated immune responses in human THP-1 macrophages and ectocervical cells. The study encompasses cell biological, immunological, and proteomic analyses to explain how Tv EVs manipulate host cell NF-κB/NLRP3 signaling pathways in macrophages and PI3K-mediated NF-κB, p38 MAPK and ERK pathways in Ect. Apart from experiments in vitro, the authors also demonstrated some of their findings in vivo. This paper provides significant contribution to our understanding of Tv EV-mediated pathogenesis and host immune response. However, there are some minor details I wish to be clarified.

**Part II – Major Issues: Key Experiments Required for Acceptance**

Reviewer #1: Fig. 2A.2B states that Tv exosomes are internalized in THP-1 cells or ectocervical cells. However, fluorescence microscopy images have low resolution and low magnification, making the cells appear small in size. Therefore, it does not accurately show evidence that Ev is internalized inside the host cell. Using cofocal microscopy, Ev internalization can be observed more clearly. Additionally, just as exosomes are stained, the cytoplasm of the host cell must also be stained. By additionally using a dye that can stain the cytoplasm, such as CtxB-Alex, you can clearly observe that Ev is internalized into the cytoplasm of the host cell. Additionally, an indication of the microscope magnification at which the observations were made is also required. It would be nice to have an enlarged photo showing the internalization of EV.

In addition, since Ev internalization into cells occurs early, it is necessary to observe the EV processing time for less than 3 hours (e.g., 30 minutes, 1 hour, 2 hours, 3 hours, etc.) to show the changing pattern. It's the same.

Reviewer #2: 1. Can the authors explain why only physical characterization of EVs amongst three Tv strains was conducted? A comparative proteomic or RNA profiling of EVs derived from three Tv isolates may provide better insights into the differences in the cytokine/chemokine responses elicited in human cells. If this analysis is performed, it may significantly improve the discussion which is currently limited to generalization as Tv EVs and not to specific proteins/RNAs directly involved in affecting host pathways.

2. The authors should also highlight and provide plausible reasons when there is differences in the responses when EVs from three different Tv strains is used.

3. Why are some experiments done not in parallel? Is there a reason why proteomic analysis after co-incubation of Tv EVs in Ect only, and not in macrophage was conducted?

**Part III – Minor Issues: Editorial and Data Presentation Modifications**

Reviewer #1: Fig. 2C, This is an electron micrograph of vesicles internalized into cells.

However, to be honest, I'm not sure if anything the authors show is clearly an Exosome. This is because macrophages can undergo exocytosis and secrete exosomes in response to Tv-EV, and cells are known to secrete exosomes. Electron micrographs appear to be mixed. Therefore, it would be good if the authors could more clearly indicate the exosomes they observed using arrows, etc. Also, the scale bar of the electron microscope photo needs to be corrected from 1um to 1μm.

Fig. 5, When treated with Ev, the p-p65 NF-kB band in THP-1 cells is too faint. And, Fig. The authors claim that NF-kB p65 moved to the nucleus in 5C, but this is difficult to confirm in micrographs. It can be clearly seen only when taken with a confocal microscope. Comparing the merged pictures, it appears that NF-kB moved to the nucleus more in the control treated group than in the TV-EV treated group. To assert the movement of NF-kB to the nucleus, it is necessary to separate the cytosol and nuclear fractions and confirm them by Western blot.

Fig. 6C and D, high-resolution photographs are required, and the movement of NF-kB to the nucleus can be observed only through confocal microscopy rather than fluorescence microscopy.

Fig. 8, TLRs (1, 2, 4, 5, 6, 10, 11) are mainly found on the cell surface, and TLRs (3, 7, 8, 9) are known to exist on the intracellular endosomal membrane. TLR3 is said to be found on the cell surface of immune cells. It is difficult to identify the distribution and location within the cell from the fluorescence microscope images shown by the authors. If we are simply showing that TLR3 expression is increased, I think Figures 8A and 8C are sufficient.

This paper shows for the first time that Tv-Ev alone is sufficient to activate the NLRP3 inflammasome in THP-1 cells. However, most experiments on inflammasome were conducted only on THP-1 cells, and inflammatory cytokine production by Tv-Ev and proteomic analysis as the last data were performed on ectocervical cells.

The relationship with the inflammasome was not proven in ectocervical cells, and the results in THP-1 cells and ectocervical cells were described unclearly in the review section. For a more complete paper, it would be better to publish the two contents separately.

Reviewer #2: 1. In Results, page 11, I do not clearly understand why the authors decide to use CD63 to test their Tv EVs. Is there Tv protein homologous to CD63? If so, please indicate in the manuscript.

2. In Results, page 13, the authors assessed the effect of Tv EVs on proliferation of host cells. They showed declined proliferation of THP-1 macrophages after 24 h incubation with Tv EVs presented in supplementary figure 1A-D. These phenotypes are also important but sufficient discussion was not included. What could have caused this phenomenon?

Minor grammatical errors:

1. Abstract: ectocervial to ectocervical

2. Results, page 11: obviously to obvious

PLOS authors have the option to publish the peer review history of their article (what does this mean? ). If published, this will include your full peer review and any attached files.

**Do you want your identity to be public for this peer review?** For information about this choice, including consent withdrawal, please see our Privacy Policy .

Reviewer #1: No

Reviewer #2: No
---

## [Decision Letter · Decision Letter 1]

PPATHOGENS-D-24-01208R1

Trichomonas vaginalis extracellular vesicles activate the NLRP3 inflammasome and TLR3-mediated inflammatory cascades in host cells

PLOS Pathogens

Dear Dr. Huang,

Thank you for submitting your manuscript to PLOS Pathogens. After careful consideration, we feel that it has merit but does not fully meet PLOS Pathogens's publication criteria as it currently stands. Therefore, we invite you to submit a revised version of the manuscript that addresses the points raised during the review process. Specifically please edit the discussion as requested by reviewer 3.

Please submit your revised manuscript within 30 days Jun 18 2025 11:59PM. If you will need more time than this to complete your revisions, please reply to this message or contact the journal office at plospathogens@plos.org. Please include the following items when submitting your revised manuscript:

We look forward to receiving your revised manuscript.

Kind regards,

Tracey J. Lamb

Section Editor

PLOS Pathogens

Tracey Lamb

Section Editor

PLOS Pathogens

Sumita Bhaduri-McIntosh

Editor-in-Chief

PLOS Pathogens

orcid.org/0000-0003-2946-9497

Michael Malim

Editor-in-Chief

PLOS Pathogens

orcid.org/0000-0002-7699-2064

**Reviewers' Comments:**

Reviewer's Responses to Questions

**Part I - Summary**

Reviewer #2: The authors have addressed the questions and comments previously given by this referee.

Reviewer #3: The authors used three strains of Trichomonas vaginalis to demonstrate that TLR3 is involved in the induction of NLRP3 inflammasomes by T. vaginalis in THP-1 cells. This revised manuscript has been reviewed by two reviewers, and the author has addressed all their questions. The manuscript has reached a fairly high level of completion. However, I hope the author can clarify a few remaining points:

**Part II – Major Issues: Key Experiments Required for Acceptance**

Reviewer #2: None

Reviewer #3: No experimental modifications are required.

**Part III – Minor Issues: Editorial and Data Presentation Modifications**

Reviewer #2: None

Reviewer #3: 1. The results presented in Figure 4 indicate that TV-EVs activate the NLRP3 inflammasome and subsequently induce IL-1β secretion in THP-1 macrophages. Previous studies have confirmed that NLRP3 can activate caspase-1 by allowing ASC to bind to pro-caspase-1 [1]. This process triggers the activation of caspase-1 and the maturation and secretion of pro-inflammatory cytokines, such as interleukin-1β (IL-1β) and IL-18. However, this study only confirmed the secretion of IL-1β in THP-1 macrophages and did not investigate IL-18. The author should discuss and explain this in the discussion section.

2. Trichomonas vaginalis inflammasome activation induces macrophage inflammatory cell death by pyroptosis, which occurs through caspase-1 cleavage of the gasdermin D protein, leading to pore formation in the host cell membrane [2]. In this manuscript, the authors demonstrated that TV-EVs induce TLR3 overexpression, leading to the activation of the NF-κB/NLRP3 pathway in THP-1 macrophages. Would the use of TV-EVs induce apoptosis or pyroptosis in in vitro experiments? Please address this question in the discussion section.

1. Malik A, Kanneganti TD. Inflammasome activation and assembly at a glance. J Cell Sci. 2017;130(23):3955-63. doi: 10.1242/jcs.207365. PubMed PMID: 29196474; PubMed Central PMCID: PMCPMC5769591.

2. Riestra AM, Valderrama JA, Patras KA, Booth SD, Quek XY, Tsai CM, et al. Trichomonas vaginalis Induces NLRP3 Inflammasome Activation and Pyroptotic Cell Death in Human Macrophages. J Innate Immun. 2019;11(1):86-98. Epub 20181102. doi: 10.1159/000493585. PubMed PMID: 30391945; PubMed Central PMCID: PMCPMC6296884.

PLOS authors have the option to publish the peer review history of their article (what does this mean? ). If published, this will include your full peer review and any attached files.

**Do you want your identity to be public for this peer review?** For information about this choice, including consent withdrawal, please see our Privacy Policy .

Reviewer #2: No

Reviewer #3: No

**Figure resubmission:**
---

## [Editor Report · Decision Letter 2]

Dear Prof. Huang,

We are pleased to inform you that your manuscript 'Trichomonas vaginalis extracellular vesicles activate the NLRP3 inflammasome and TLR3-mediated inflammatory cascades in host cells' has been provisionally accepted for publication in PLOS Pathogens.

Best regards,

Tracey J. Lamb

Section Editor

PLOS Pathogens

Tracey Lamb

Section Editor

PLOS Pathogens

Sumita Bhaduri-McIntosh

Editor-in-Chief

PLOS Pathogens

orcid.org/0000-0003-2946-9497

Michael Malim

Editor-in-Chief

PLOS Pathogens

orcid.org/0000-0002-7699-2064
---

## [Editor Report · Acceptance letter]

Dear Prof. Huang,

We are delighted to inform you that your manuscript, "Trichomonas vaginalis extracellular vesicles activate the NLRP3 inflammasome and TLR3-mediated inflammatory cascades in host cells," has been formally accepted for publication in PLOS Pathogens.

Best regards,

Sumita Bhaduri-McIntosh

Editor-in-Chief

PLOS Pathogens

orcid.org/0000-0003-2946-9497

Michael Malim

Editor-in-Chief

PLOS Pathogens

orcid.org/0000-0002-7699-2064